# Role of precipitation and extreme precipitation events on the variability of ice core surface mass balances in Dronning Maud Land: insights from RACMO2.3 and statistical downscaling

Sarah Wauthy[1,2] and Quentin Dalaiden[3,4]

[1]Laboratoire de Glaciologie, Département Géosciences, Environnement et Société (DGES), Université libre de Bruxelles (ULB), Brussels, Belgium
[2]Laboratoire G-Time, Département Géosciences, Environnement et Société (DGES), Université Libre de Bruxelles (ULB), Brussels, Belgium
[3]Earth and Life Institute (ELI), Université catholique de Louvain (UCLouvain), Louvain-la-Neuve, Belgium
[4]Nansen Environmental and Remote Sensing Center (NERSC) and Bjerknes Center for Climate Research, Bergen, Norway

*Correspondence to*: Sarah Wauthy (sarah.wauthy@ulb.be)

**Abstract.** The Antarctic Ice Sheet (AIS) is the most uncertain contributor to future sea level rise for projections by the end of this century. One of the main drivers of future AIS mass changes is the surface mass balance (SMB) of the ice sheet, which is associated with a number of uncertainties, including its large temporal and spatial variability. The SMB is influenced by a complex interplay of the various processes driving it, including large-scale atmospheric circulation, ice sheet topography, and other interactions between the atmosphere and the snow/ice surface. This spatial and temporal variability is identified in three ice cores located at the crests of adjacent ice rises in coastal Dronning Maud Land, each approximately 90 km apart, which show very contrasting SMB records. In this study, we analyze the role of precipitation and extreme precipitation events (EPEs) in this variability. Our results, based on the RACMO2.3 model and a dataset derived from statistical downscaling, confirm that precipitation is the primary driver of SMB, and that synoptic-scale EPEs play a significant role in controlling interannual variability in precipitation and thus SMB. Shedding light on the intricate nature of SMB variability, our results also demonstrate that precipitation and EPEs alone cannot explain the observed contrasts in SMB records among the three ice core sites and suggest that other processes may be at play. This underscores the importance of adopting comprehensive, interdisciplinary methods, like data assimilation that combines observations and the physics of models, to unravel the underlying mechanisms driving this variability.

## 1 Introduction

The Antarctic Ice Sheet (AIS) has contributed to about 10 % of the sea level rise observed between 2006 and 2018 and has been losing mass at an accelerating rate until 2016 (Fox-Kemper et al., 2021). Since 2016, the mass loss has not continued to increase due to regional mass gains at the surface, particularly in Dronning Maud Land (Velicogna et al., 2020). A recent study by Wang et al. (2023) reported a total mass gain of 130 Gt yr$^{-1}$ between 2021 and 2022, setting a record for the satellite period. This mass gain, observed over the East AIS and the Antarctic Peninsula, exceeded the mass loss in the Amundsen sector of the West AIS, which is driven by the intrusions of warm water masses beneath the ice shelves. For each degree of warming, a precipitation increase of 7 % should be observed (Krinner et al., 2006; Palerme et al., 2014; Frieler et al., 2015; Dalaiden et al., 2020). The projected snow accumulation increase in the coming decades might therefore play a pivotal role in mitigating the basal mass loss. Examining surface processes is key to understanding why the ice sheet has gained surface mass in the past years. With an equivalent of 58.0 m of sea level rise (Oppenheimer et al., 2019), the AIS is the largest potential contributor and yet the most uncertain one to the future sea level rise (Kopp et al., 2017). The three main drivers of this future AIS change – sub-shelf melting, ice shelf disintegration, and surface mass balance – are all expected to increase under a warming climate (Fox-Kemper et al., 2021).

Surface mass balance (SMB) in Antarctica is largely dominated by the amount of solid precipitation (van Wessem et al., 2018; Agosta et al., 2019; Lenaerts et al., 2019). The effect of higher temperature, referred to as thermodynamics, is one of the three mechanisms controlling precipitation variability (Dalaiden et al., 2020). Albeit interconnected, the two other mechanisms are related to atmospheric dynamics, which can be further separated into two spatiotemporal scales: large-scale dynamics and synoptic-scale dynamics (Dalaiden et al., 2020). Large-scale dynamics correspond to the southward moisture transport from lower latitudes (Marshall et al., 2017), while short-lived intrusion of maritime air resulting in high precipitation are generally related to synoptic-scale dynamics. Such short-lived intrusions correspond to extreme precipitation events (EPEs) which can be associated with atmospheric rivers (Turner et al., 2019). Atmospheric rivers are defined by the American Meteorological Society as "a long, narrow, and transient corridor of strong horizontal water vapor transport that is typically associated with a low-level jet stream ahead of the cold front of an extratropical cyclone".

Dynamical changes induce a strong regional variability due to the interactions of winds with the topography. This for instance explains the high rate of snow accumulation over the eastern part of the Antarctic Peninsula (due to adiabatic cooling; windward side) and the low rate on the leeward side in the western Peninsula (van Wessem et al., 2016). Similarly, ice rises along the Antarctic coastline significantly influence the spatial variability in precipitation, and hence accumulation, over smaller scales, typically spanning a few kilometers (Lenaerts et al., 2014). The orographic uplift on the windward side of the ice rise enhances precipitation, while snow erosion takes place on the leeward side, leading to a local gradient of the SMB distribution across the ice rise (Lenaerts et al., 2014). In the Dronning Maud Land (DML) coastal region, on the Lokeryggen ice rise, a significant contrast between the windward and the leeward sides of the ice rise was observed with a SMB up to 1.5 times higher on the windward side over a 30 year-period (Kausch et al., 2020).

Beyond spatial variability, SMB is also characterized by strong temporal variability. Medley and Thomas (2019) identified an Antarctic-wide snow accumulation increase between 1801 and 2000, which mitigated the 20th century sea level rise by approximately 10 mm. However, these snow accumulation reconstructions strongly vary, both in sign and magnitude, at the regional scale (Medley and Thomas, 2019). While the long-term and large-scale snow accumulation changes are well understood in West Antarctica (e.g., Medley and Thomas, 2019; Dalaiden et al., 2021), the variability in East Antarctica is more complex. Medley and Thomas (2019) found a general increase over the East AIS between 1801 and 2000 but a decrease in the late 20th century, along with strong local trends – some positive, others negative – during the second half of the 20th century. Most of the ice cores in coastal DML show a decreasing SMB trend over recent decades (Kaczmarska et al., 2004; Sinisalo et al., 2013; Schlosser et al., 2014; Altnau et al., 2015; Vega et al., 2016; Ejaz et al., 2021) but an ice core shows a significant increasing trend from mid-20th century (Philippe et al., 2016). A recent study added two new records to the region: one shows a significant decreasing trend, while the other presents a more complex pattern with alternating periods of increases and decreases (Wauthy et al., 2024).

These two new records, along with the one of Philippe et al. (2016), are located at the crests of three adjacent ice rises, each approximately 90 km apart. These records capture significant SMB variability making them ideal locations to study the spatial and temporal variability. The complexity of Antarctic SMB, as well as its historical and contemporary changes, highlights the crucial need for a deeper understanding of the processes involved. Such knowledge is essential for improving our ability to predict the future contribution of Antarctic snow accumulation to global sea-level rise.

In this paper, we aim to understand the SMB temporal variability observed at the three ice rises mentioned above and to decipher the processes driving this spatiotemporal variability. To this end, the spatial and temporal variability of precipitation

and extreme precipitation events are analyzed using a state-of-the-art polar-oriented regional climate model, RACMO2.3, over the satellite era and a recent dataset providing an ensemble of simulations at high spatial resolution since 1850. The synoptic conditions associated with the EPEs are studied, as well as the co-occurrence of precipitation anomalies.

## 2 Data and methods

### 2.1 Ice cores SMB

Three ice cores have been drilled on adjacent ice rises along the Princess Ragnhild coast in DML (see Fig. 1): Derwael ice rise (DIR), Lokeryggen ice rise (LIR), and Hammarryggen ice rise (HIR). The westernmost (HIR) and easternmost (DIR) ice rises are approximately 180 km apart. Note that, from a geomorphological point of view, the Lokeryggen and Hammarryggen ice rises are ice promontories connected to the grounded ice sheet to the south and surrounded by ice shelves to the east, north and west. The IC12 core was drilled in December 2012 at the crest of DIR (−70.24218 °S, 26.34162 °E; ~429 m ASL) and is 120 m long. The FK17 core was drilled during the 2017/2018 austral summer at the crest of LIR (-70.53648° S, 24.07036° E; ~333 m ASL) and is 208 m long. The TIR18 core was drilled during the 2018/2019 austral summer at the crest of HIR (-70.49960° S, 21.88017° E; ~348 m ASL) and is 262 m long. Only the top 120 m of both FK17 and TIR18 cores have been analyzed (Wauthy et al., 2024).

The three ice cores were dated to the end of the eighteenth century at a depth of 120 m (CE 1759 ± 16 years, CE 1793 ± 3 years and 1780 ± 5 years, for IC12, FK17, and TIR18 respectively). Annual layers have been identified using water stable isotopes ($\delta^{18}$O, $\delta$D), specific major ions with seasonal signal and electrical conductivity measurement with the identification of volcanic horizons allowing to refine the relative dating. The complete dating procedures are described in Philippe et al. (2016) for IC12 and Wauthy et al. (2024) for FK17 and TIR18. The water equivalent annual layer thicknesses are obtained from combining raw annual layer thicknesses with density profiles. Subsequently, these thicknesses are corrected to account for the thinning effects of divergent ice flow at ice rises.

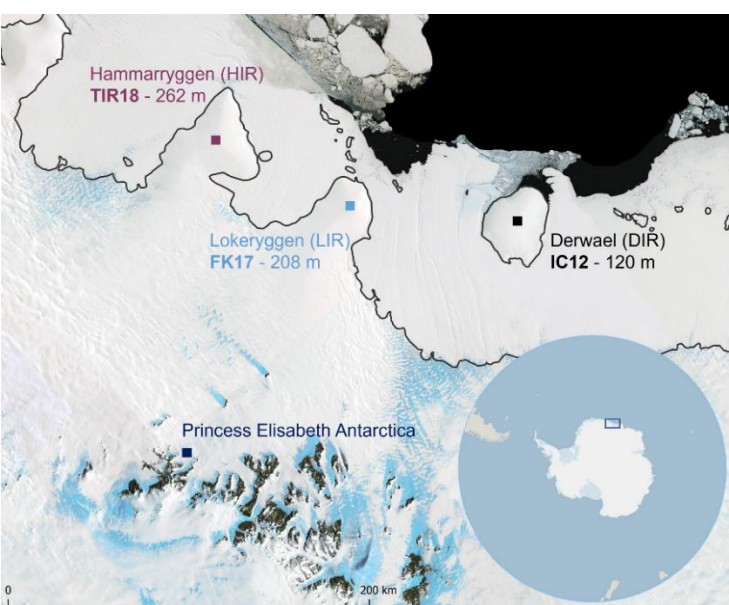

**Figure 1: Location of the three ice cores drilled at the crest of the ice rises, from west to east: Hammarryggen (HIR), Lokeryggen (LIR) and Derwael (DIR). Region of the Princess Ragnhild Coast, Dronning Maud Land, East Antarctica. Figure from Wauthy et al. (2024).**

The mean SMB over 1816–2011 is comparable for IC12, FK17, and TIR18, with respectively 477 mm w.e. yr⁻¹, 532 mm w.e. yr⁻¹, and 504 mm w.e. yr⁻¹. However, the three annual SMB records exhibit pronounced differences regarding the temporal

variability (Fig. 2). The IC12 core shows relatively stable SMB from 1750 to 1950, with intermittent interdecadal variability,
followed by a significant increase until the end of the record in 2011. In contrast, FK17 presents a more complex pattern with
long-term oscillations: an increase from 1793 to about 1825, a decrease until ~1925, another increase and plateau until about
1995, and a recent decreasing trend. TIR18 displays higher variability with no statistically detectable trend before 1850 but
exhibits a significant SMB decrease extending to the end of the record in 2018.

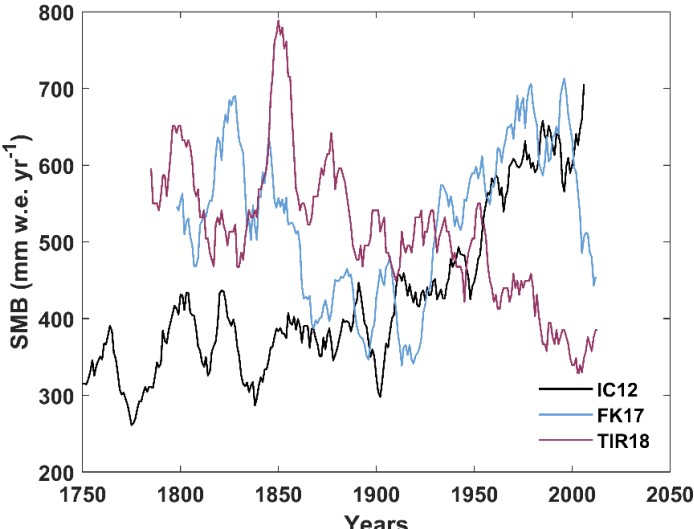

**Figure 2: Surface mass balance records from the three ice cores, expressed in millimeter water equivalent per year (IC12 in black,**
**FK17 in blue and TIR18 in burgundy). These have been corrected for vertical strain rates and smoothed using an 11-year running**
**mean. Figure adapted from Wauthy et al. (2024).**

**2.2 The Regional Atmospheric Climate MOdel (RACMO)**

The SMB can be described as in Lenaerts et al. (2012):

$$SMB = P - SU_s - SU_{ds} - ER_{ds} - RU ,$$ (1)

where P corresponds to the total precipitation (snowfall and rain), $SU_s$ represents the surface loss by sublimation and by
evaporation, $SU_{ds}$ is the drifting snow sublimation and $ER_{ds}$ corresponds to the drifting snow erosion/deposition – drifting
snow is caused by the near-surface winds (i.e., blowing snow) –, and RU is the meltwater runoff.

We use a regional climate model (RCM) able to accurately simulate surface processes over ice sheets, more specifically the
Regional Atmospheric Climate MOdel version 2.3p2, referred to as RACMO2.3 here. RACMO2.3 is specifically applied to
the polar regions (van Wessem et al., 2018), including a specific regional simulation centered on DML at 5.5 km horizontal
resolution (Lenaerts et al., 2014) spanning the 1979–2016 period. This simulation used the ERA-Interim atmospheric
reanalysis (Dee et al., 2011) as forcing at its lateral boundaries. To analyze the precipitation component of the SMB at a daily
resolution, we use the total precipitation provided by the RACMO2.3 grid cells corresponding to each of the three ice core
sites. The grid cells are named following to the ice core sites: IC for IC12 site, FK for FK17 site, and TIR for TIR18 site. A
precipitation day is defined as a day with more than 0.02 mm of precipitation (Turner et al., 2019).

**2.3 Downscaling**

Considering the strong internal variability of the Antarctic climate system (e.g., Jones et al., 2016; King and Watson, 2020),
the satellite period might be too short to study the long-term variability of the SMB. An opportunity to analyze longer periods
rises from statistical downscaling which extends the time period covered by the RACMO2.3 simulation. Here, we use the
dataset from Ghilain et al. (2022) that employed a statistical downscaling method combining the daily snowfall from an RCM
at high spatial resolution to specific weather patterns associated with snowfall events in state-of-the-art atmospheric reanalyses.

Based on these dynamical relationships, Ghilain et al. (2022) applied this relationship on an ensemble of simulations performed with an Earth System Model (ESM) covering the 1850–2014 period to downscale precipitation over the DML coastal region at a high resolution of 5.5 km (see the original article for details and specifications). Briefly, this dataset provides daily snowfall by using the RACMO2.3 simulation at 5.5 km horizontal resolution (Lenaerts et al., 2014), ERA5 as the atmospheric reanalysis for the weather patterns (Hersbach et al., 2020), and an ensemble of 10 simulations performed with the Community Earth System Model version 2 (CESM2; Danabasoglu et al., 2020) at low horizontal resolution (1 degree). The resulting 10 daily snowfall simulations combine the advantages of the polar-oriented surface physics from the RCM along with the weather patterns from state-of-the-art atmospheric reanalysis as well as the long time period covered by the ESM (i.e., 1850-2014).

## 3 Results

### 3.1 Comparison between ice core observations and models

Major differences are observed between the SMB simulated by the models (plain lines, Fig. 3) and the ice-core records (dashed lines, Fig. 3). From 1979 to 2016 (left panels – Fig. 3), RACMO2.3 consistently underestimates SMB compared to ice-core observations, with differences of approximately 200 mm for both FK (36 % of the ice core SMB) and TIR (49 %), and about 265 mm for IC (44 %). The correlation coefficients between the ice cores and RACMO2.3 are relatively low, with values of 0.17, 0.40, and 0.65 for FK, TIR, and IC, respectively. Additionally, RACMO2.3 tends to underestimate the magnitude of the interannual variability: its standard deviation ranges from 77 to 103 mm, compared to 135 to 199 mm in the ice-core records. Although RACMO2.3 has a spatial resolution of 5.5 km, it may still be too coarse to accurately reproduce the magnitude of interannual variations from the ice-core records.

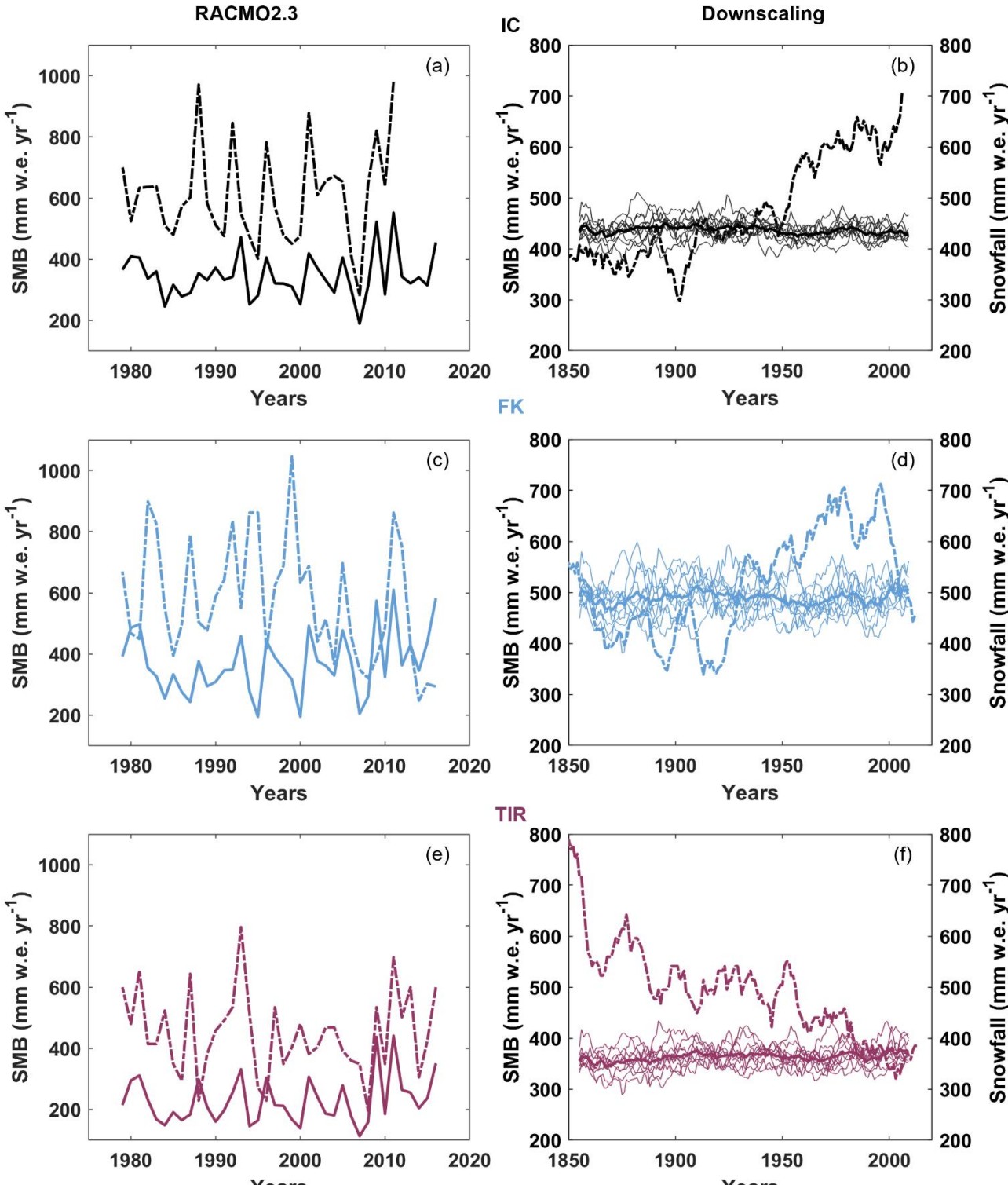

**Figure 3: Comparison between ice core observations and models at the three sites: IC (in black), FK (in blue), and TIR (in burgundy). The left panels represent the SMB from the ice core records (dashed lines) and the SMB from RACMO2.3 (plain lines) between 1979 and 2016. The right panels represent the SMB from the ice core records (dashed lines) and the annual snowfall of the 10 members (plain lines) and average of the 10 members (thick plain line) from the downscaling between 1850 and 2014. For visibility reasons, the ice core and downscaling datasets have been smoothed using an 11-year running mean.**

For the 1850-2014 period, only daily snowfall ensembles, considered as an overestimation of SMB, are available from the statistical downscaling. The right panels of Fig. 3 show annual snowfall data from the 10 ensemble members, the ensemble snowfall average, and ice core SMB records for each site. It is evident that the downscaling dataset is not capturing the long-term trends of the ice core records. However, the average values for the 1850-2014 period are closer to those of the ice core SMBs than RACMO2.3, with average differences inferior to 50 mm for FK and IC (due to both under and overestimation).

For TIR, the downscaling dataset appears to consistently underestimate the ice-core observations before 1980 (129 mm). With standard deviation ranging from 66 to 108 mm, interannual variability is also lower in the downscaling dataset than in the ice core records. When comparing snowfall from RACMO2.3 and downscaling dataset (not shown) on the 1979-2014 period, it appears that average snowfall is similar, with differences inferior to 5 mm for IC and FK, and to 14 mm for TIR. The internal variability is also comparable for the three sites. This is expected as the downscaling dataset is based on RACMO2.3 at 5.5 km horizontal resolution.

These differences will be thoroughly discussed in the Discussion. However, it is worth mentioning that, despite the existing room for improvements, both RACMO2.3 at 5.5 km and downscaling datasets are the best tools available for this study, as their finer spatial scale can capture more realistic SMB. Indeed, the 5.5 km scale RACMO2.3 has been well evaluated in our region of interest (Lenaerts et al., 2014; Kausch et al., 2020; Ghilain et al., 2022) and is expected to better represent the atmospheric and surface dynamics than RACMO2.3 at a 27 km horizontal resolution. Regarding the downscaling dataset, Ghilain et al. (2022) observed a general good agreement between the downscaled snowfall and eight ice cores located in the DML coastal region and highlight an important bias reduction in comparison to CESM2 without downscaling. Furthermore, ice core records are complex, with variable SMB trends observed in multiple ice core records from coastal DML. But they are also subject to biases, particularly the influence of non-climatic factors.

We thus make the hypothesis that RACMO2.3 and the statistical downscaling perform well enough to investigate the potential processes explaining the large variability observed. Since RACMO2.3 spans a short period of time, the downscaling dataset is used to increase the sample size and extend the record.

It should be noted that the differences between precipitation and snowfall are negligible at the three sites studied (~ 0.1 mm yr$^{-1}$) and that the terms will therefore be used interchangeably from here on.

## 3.2 Contribution of the SMB components in RACMO2.3

To investigate the drivers of SMB variability, we analyze its components (Eq. 1) using RACMO2.3. Meltwater runoff is negligible, particularly on the grounded ice sheet (van Wessem et al., 2018), so we focus on the four other components: total precipitation (P), surface loss by sublimation ($SU_s$), sublimation ($SU_{ds}$) and erosion/deposition ($ER_{ds}$) caused by drifting snow. Figure S1 in the Supplement illustrates the temporal evolution of these components from 1979 to 2016, highlighting their relative contributions to total SMB, for each site.

As expected, precipitation is the main component of the annual mean SMB. For both the IC and FK sites, the erosion by the blowing snow also plays a major role in the mean SMB with a contribution reaching 23.0 % and 34.6 % of the total mean modelled SMB, respectively. In addition, the SMB at the TIR site is governed by the blowing snow-related sublimation (contribution of 19.2 %), which indicates that this site is highly prone to blowing snow. In contrast, the SMB at the FK site is less impacted by the blowing snow, with a smaller contribution to the mean modelled SMB from the erosion (13.2 %) and sublimation (17.1 %).

The annual SMB variability closely follows year-to-year precipitation at all three sites, with anomalously high/low SMB corresponding to anomalously high/low precipitation (Fig. S1 in the Supplement). Precipitation accounts for over 90 % of the SMB variance for each site, with negligible contributions from other components. Variance is a measure of variability calculated by taking the squared standard deviation. Thus, according to RACMO2.3, SMB temporal variability is primarily driven by precipitation.

### 3.3 Precipitation: spatial variability and temporal trends

Precipitation is the key driver of the SMB variability at our three ice core sites according to RACMO2.3, we thus analyze the annual total precipitation variability over 1979–2016 using RACMO2.3 and the high-resolution precipitation downscaling dataset from Ghilain et al. (2022) to study the variability since 1850.

According to RACMO2.3, precipitation occurs on approximately half of the days during the 1979–2016 period (54 % for FK and TIR, and 62 % for IC). To better understand the spatial distribution of snowfall between sites, the distribution of the precipitation days is examined (Table 1). Specifically, when precipitation occurs at one site, the conditions at the two other sites are evaluated. This approach helps determine whether a precipitation day at one site coincides with dry conditions at another, potentially explaining the contrasting signals in the ice-core records (Fig. 2). The analysis considers both the number of events and the precipitation quantities: for each site, the proportion of total precipitation occurring during events affecting all three sites, only the specific site, or the site and one other location is calculated.

| | Number of events (%) | | | | Quantity (%) | | | |
|---|---|---|---|---|---|---|---|---|
| | all sites | site only | + site 1 | + site 2 | all sites | site only | + site 1 | + site 2 |
| IC | 75.6 | 13.3 | 7.7 (FK) | 3.4 (TIR) | 95.7 | 1.3 | 2.5 (FK) | 0.5 (TIR) |
| FK | 86.4 | 1.4 | 8.9 (IC) | 3.3 (TIR) | 99.1 | 0.1 | 0.6 (IC) | 0.2 (TIR) |
| TIR | 87.2 | 5.5 | 4.0 (IC) | 3.3 (FK) | 98.4 | 0.5 | 0.3 (IC) | 0.8 (FK) |

**Table 1: Distribution of precipitation days at the three sites using RACMO2.3. An event occurring at one site is categorized differently if it affects all three sites, the considered site only or the considered site and one of the two other sites (in this case, the name in parenthesis indicates which site is the second site impacted). The "number of events" represents the fraction of the total number of events in the category. Quantity represents the percentage of total precipitation that fell at one site based on the event category.**

It appears that a significant proportion of precipitation events affects all three sites simultaneously, though IC experiences more events occurring only at this site. However, this characteristic only represents 1.3 % of IC's total precipitation. For all three sites, over 95 % of total precipitation occurs during events affecting the three sites simultaneously. Furthermore, high correlations in annual total precipitation time-series among the sites are observed (Fig. S2 in the Supplement), with correlation coefficients greater than 0.85 (0.88 for FK-IC correlation, 0.91 for TIR-IC, and 0.93 for TIR-FK). This means that the temporal variability of the precipitation at these three sites is highly similar. Therefore, RACMO2.3 suggests minimal spatial variability in precipitation across the three sites.

To examine precipitation changes over longer periods, daily precipitation time-series from the downscaling dataset are also analyzed (Table S1 and Fig. S3 in the Supplement). During the period from 1850 to 2014, over 85% of days are defined as precipitation days (86 % for FK and TIR, and 88 % for IC). This is much higher than in RACMO2.3. However, similar to RACMO2.3, the distribution of the precipitation days in Table S1 indicates that most events (nearly 80%) occur at all three sites at the same time, which corresponds to about 97 % of total precipitation. The correlation between the annual total precipitation time-series of the three sites is examined for each of the 10-member ensemble (Fig. S3). High correlations are found, with an average correlation coefficient of 0.82 (0.80 for FK-IC correlation, 0.78 for TIR-IC, and 0.87 for TIR-FK). Overall, the results align with those from RACMO2.3 about the precipitation distribution for the satellite era, confirming the absence of spatial variability in annual precipitation across the three sites, even over longer time scales.

Potential long-term trends and multidecadal variability are examined using the annual downscaling snowfall data (right panels in Fig. 3). The results reveal significant multi-decadal variability at all sites but no clear long-term trends, in contrast with ice-core records. The distribution of precipitation anomalies (fluctuations along the mean) at the annual resolution in both outputs

of RACMO2.3 and the 10 members from the downscaling is summarized in Table 2. Overall, Table 2 confirms concurrent annual SMB anomalies across the three sites. For both datasets, the three sites show the same anomaly sign (either positive or negative) in over 70 % of cases. FK and TIR are slightly more closely connected, sharing the same anomaly sign more frequently, while IC shows an opposite sign in about 13 % of cases. The remaining cases account for 5–10 %.

| Distribution of precipitation anomalies (%) | +IC +FK +TIR | -IC -FK -TIR | +IC +FK -TIR | -IC -FK +TIR | +IC -FK +TIR | -IC +FK -TIR | -IC +FK +TIR | +IC -FK -TIR |
|---|---|---|---|---|---|---|---|---|
| RACMO2.3 | 31.6 | 39.5 | 0 | 5.3 | 2.6 | 7.9 | 5.3 | 7.9 |
| Downscaling | 32 ± 4 | 40 ± 4 | 4 ± 2 | 4 ± 2 | 3 ± 2 | 4 ± 1 | 6 ± 2 | 7 ± 2 |

**Table 2: Distribution of precipitation anomalies (% of total cases) at the annual resolution in both outputs of RACMO2.3 and the 10-member average from the downscaling. The "+" sign corresponds to positive anomalies, and the "-" sign corresponds to negative anomalies. For the downscaling, the ± standard deviation values represent the variability between the members.**

### 3.4 Extreme Precipitation Events

In addition to playing a role in SMB variability, Extreme Precipitation Events (EPEs) have been shown to significantly contribute to the total annual precipitation across Antarctica (Turner et al., 2019). Even though there is not a unique definition of an EPE in the literature (Turner et al., 2019 and references therein), an EPE day is defined here as a day when the total precipitation exceeds a certain percentile value of the total daily precipitation distribution. Here, we use the 95th and the 98th percentiles. The 98th percentile corresponds to the largest 2% of daily precipitation events, of which approximately half are caused by the intense localized snowfall events produced by atmospheric rivers (Wille et al., 2021). EPE threshold values are calculated independently for each model grid cell corresponding to each ice core site, for both percentiles, and for RACMO2.3 and downscaling datasets. EPEs are analyzed to investigate potential spatial and/or temporal trends that might explain the different SMB trends observed in the three ice cores. This analysis is applied to precipitation days from both the RACMO2.3 (1979–2016) and downscaling (1850–2014) datasets.

### 3.4.1 Contribution of EPEs to the total annual precipitation and to its variability

Annual EPE precipitation is defined as the sum of precipitation during EPEs for each year (mid and dark colors, Fig. 4). For the downscaling dataset, annual EPE precipitation time-series are derived for each of the 10 members, and their average is calculated for the three sites. The averages of the 10 members for the 95th percentile and 98th percentile, and the total precipitation are plotted for IC, FK, and TIR in Fig. 4b, 4d, and 4f, respectively.

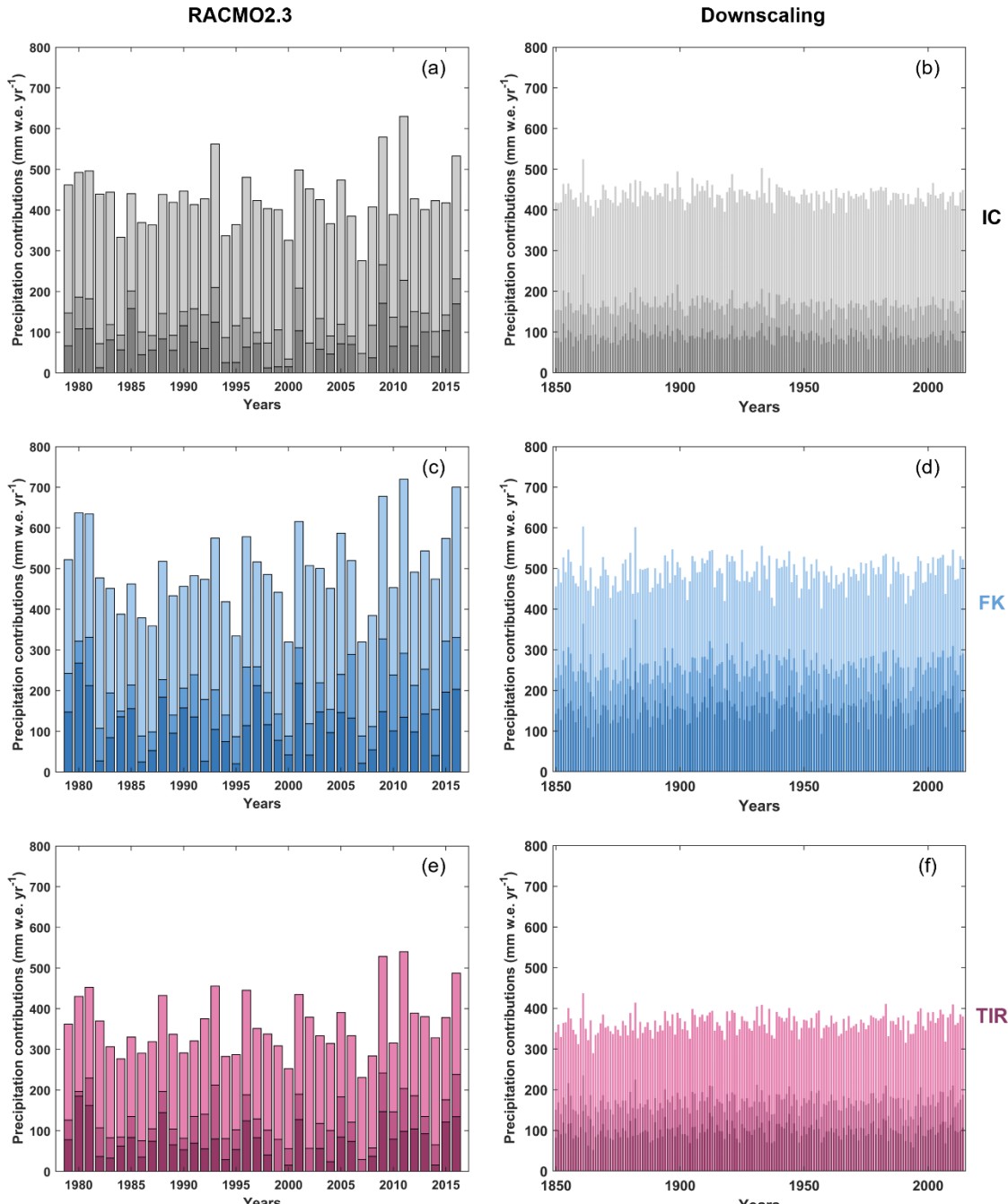

**Figure 4: Contributions of the EPEs defined by the 95th and 98th percentiles to the total annual precipitations (RACMO2.3; left panels) and to the total annual precipitations (downscaling; right panels) at the three sites: (a-b) IC, (c-d) FK, and (e-f) TIR. The contribution of the 98th EPEs is represented by the dark color, the contribution of the 95th EPEs corresponds to the sum of the dark and mid colors (since the 98th EPEs are included in the 95th EPEs), and the total annual precipitation is the sum of the dark, mid, and light colors. Note that for visibility reasons, the standard deviation (representing the internal variability) between the 10 members of the downscaling dataset is not shown here.**

The contributions of EPEs to the total annual precipitation vary in space and time (Table 3) in both RACMO2.3 (Fig. 4a, 4c, and 4e) and downscaling datasets (Fig. 4b, 4d, and 4f).

| | 95th percentile | | | 98th percentile | | |
|---|---|---|---|---|---|---|
| | IC | FK | TIR | IC | FK | TIR |
| RACMO2.3 | 30.0 | 40.0 | 35.2 | 15.6 | 22.4 | 19.1 |

| | | | | | | | |
|---|---|---|---|---|---|---|---|
| Average contribution (%) | Downscaling | 37 ± 8 | 51 ± 9 | 46 ± 9 | 20 ± 7 | 30 ± 11 | 26 ± 10 |
| Variance (%) | RACMO2.3 | 50.8 | 57.5 | 54.2 | 35.4 | 38.9 | 36.3 |
| | Downscaling | 68 ± 5 | 83 ± 5 | 78 ± 5 | 42 ± 6 | 60 ± 6 | 53 ± 7 |

**Table 3: Average contribution (upper rows) and variance (lower rows) of EPEs to the total annual precipitations from RACMO2.3 and statistical downscaling for the three sites, expressed in %. For the downscaling, the ± standard deviation values represent the internal variability of the ensemble.**

Table 3 illustrates that EPEs consistently contribute more to the total annual precipitation at FK than at the other sites. The EPE variance accounts for more than one-third to more than two-thirds of the SMB variance. This confirms the critical role of EPEs in the SMB variability. Major differences are observed between both datasets. The average contribution and variance are substantially higher in the downscaling dataset than in RACMO2.3. To investigate this, the average number of EPEs per year was calculated for both datasets and both percentiles. For the downscaling dataset, overall, all sites are characterized by the same average number of EPEs per year (16 for the 95[th] percentile, and 6 for the 98[th] percentile). For RACMO2.3, the number of events is similar across the three sites, with one additional event occurring at IC compared to the other two sites (95[th] percentile: 10 for FK and TIR, 11 for IC; 98[th] percentile: 4 for FK and TIR, and 5 for IC). Although these values indicate no discernible spatial variability between the three sites in terms of the number of events per year, there are significantly more events according to downscaling than RACMO2.3, particularly for the 95[th] percentile. These observations suggest that the downscaling product might be biased towards higher precipitations, thus overexpressing these events, because of the approach used for generating the dataset – i.e., by focusing on the precipitation events. However, this does not mean that downscaling is significantly less effective than RACMO2.3 at capturing spatial variability.

As for annual precipitation time-series, correlations are evaluated between the annual EPE precipitation of the three sites. The resulting correlation coefficients are high for both percentiles, indicating a strong relationship of the precipitation variability at the three sites. This is especially evident in RACMO2.3 annual EPE precipitation time-series, with an average correlation coefficient of 0.79 for the 95[th] percentile (0.74 for FK-IC correlation, 0.82 for TIR-IC and 0.82 for TIR-FK). Albeit the correlation still remains high when considering the 98[th] percentile, we notice a weaker relationship (0.69 for FK-IC, 0.75 for TIR-IC, and 0.80 for TIR-FK). The correlation coefficients are lower in the downscaling dataset, with an average coefficient of 0.75 for the 95[th] percentile (0.73 for FK-IC, 0.70 for TIR-IC, and 0.82 for TIR-FK) and 0.68 for the 98[th] percentile (0.65 for FK-IC, 0.62 for TIR-IC, and 0.77 for TIR-FK). The resulting weaker relationship when considering extreme events compared to annual time-series suggests that EPE impacts are more localized compared to the average conditions. Similar to the total annual precipitation time-series, the annual EPE precipitation time-series from RACMO2.3 show no significant temporal trend at any site for both percentiles. A similar conclusion can be drawn from the downscaling dataset, although one to two ensemble members are characterized by significant trends (not shown), no general pattern has been identified. The number of EPE days also shows no significant temporal trends either (not shown). In summary, EPEs play a major role in SMB variability but no significant temporal trends are observed in the EPEs contributions to the total annual precipitations, at all sites, for the two EPEs thresholds, and in both RACMO2.3 and downscaling datasets. Consequently, this rules out EPEs as the cause of the different SMB trends observed in the three ice cores. Nonetheless, the analysis of EPEs indicates that a stronger variability at the spatial scale associated with extreme events is expected.

### 3.4.2 Synoptic patterns during EPEs

Synoptic conditions during EPEs are investigated using the daily sea-level pressure (SLP) fields at 700 hPa and wind fields from ERA5 atmospheric reanalysis at the dates corresponding to EPEs according to RACMO2.3. ERA5 is expected to capture

similar EPEs to those of RACMO2.3, since the latter uses a previous version of the reanalysis (ERA-Interim) for the boundary conditions, and with the modest size of the domain, the atmospheric dynamics are relatively similar, although differences remain between RACMO2.3 forced by ERA-Interim and RACMO2.3 forced by ERA5 (e.g., Carter et al., 2022). Figure 5 presents the mean daily sea-level pressure anomalies and mean surface wind vectors (relative to the 1979–2022 monthly climatology) for all 95[th] percentile EPEs at the three sites. The synoptic conditions reveal a large negative sea-level pressure anomaly centered over Dronning Maud Land, with an associated high sea-level pressure located in the East of Droning Maud Land. This dipole structure forms a blocking ridge over the region east of the ice rises and brings a large quantity of moisture from the lower latitudes to the ice core sites. This is a typical synoptic pattern during EPEs (Turner et al., 2019; Wille et al., 2021; Simon et al., 2024). The large-scale synoptic pattern shows no significant differences across the three sites. Consistent with the sea-level pressure patterns, the surface wind fields during these events show predominantly easterly and southward flow over the region encompassing the three coastal sites. Similar observations are made for EPEs defined with the 98[th] percentile (see Fig. S4 in the Supplement). According to RACMO2.3 along with ERA5, the synoptic conditions associated with the EPEs therefore cannot explain the spatial variability observed from ice-core records.

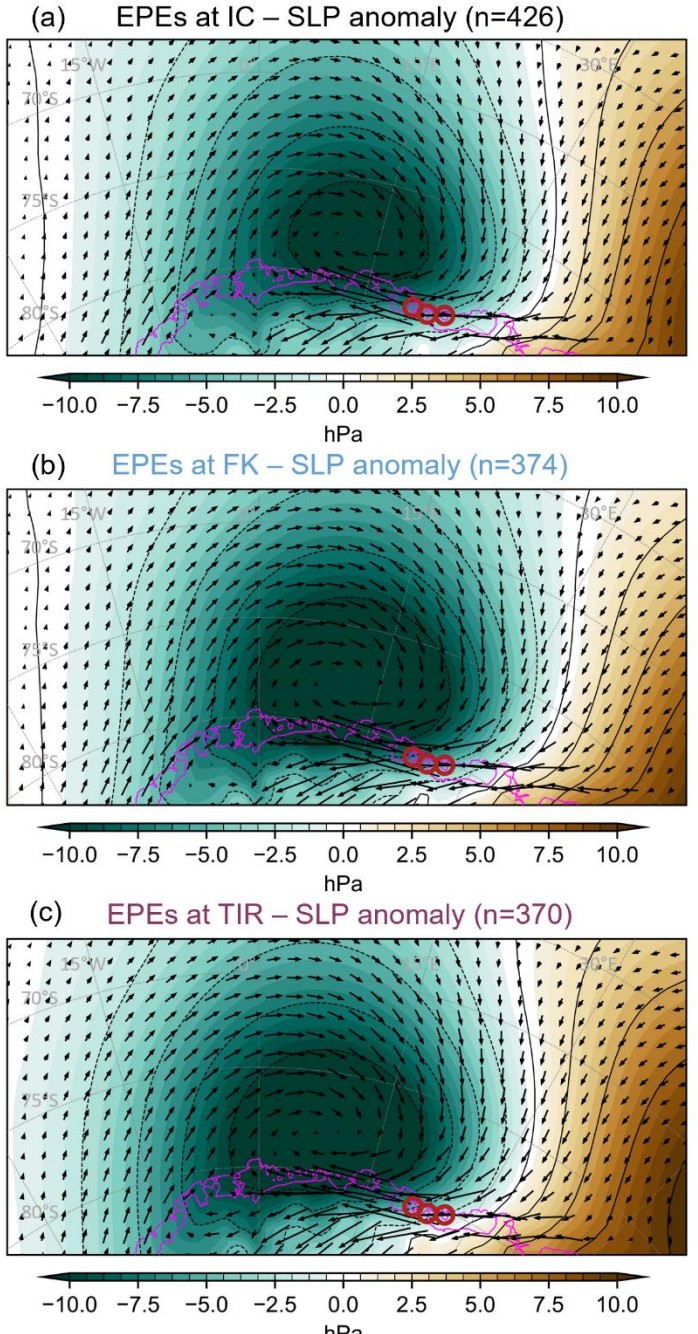

**Figure 5:** Maps of the mean daily sea-level pressure anomaly (SLP) overlaid by mean surface wind vectors, retrieved from ERA5, for all 95th percentile EPEs at the three sites: (a) IC, (b) FK, and (c) TIR. The number in parentheses corresponds to the number of EPE days. Blue colors indicate negative anomaly (i.e., low pressure) and brown colors indicate positive anomaly (i.e., high pressure). The anomalies are relative to monthly climatology calculated over the 1979–2022 period. The location of the three ice cores is shown as circles.

### 3.5 Co-occurrence of precipitation anomalies

A particularly high precipitation event at one site might result in abnormally low precipitation at the two other sites, possibly explaining the different SMB trends observed in the three ice cores. It is reasonable to hypothesize that as the primary moisture in the air is released over the first site (in form of snow), i.e., when the air mass is orographically lifted, the remaining air holds less humidity, leading to drier conditions and reduced precipitation at downstream sites (along the wind flow). To explore this, we study the frequency distributions of precipitation anomalies at the two other sites on dates corresponding to EPEs at the first site. Precipitation anomalies are calculated for each site by subtracting the average value of precipitation days from each daily precipitation value. Results are shown here for the 95th percentile EPEs, and in Sect. S5 in the Supplement for the 98th percentile EPEs.

### 3.5.1 Frequency distributions of precipitation anomalies

The frequency distributions of precipitation anomalies are presented in Fig. 6, with RACMO2.3 results on the left panels and downscaling results on the right. When an EPE occurs at one site, the two other sites may receive either below-average precipitation (negative anomalies), more precipitation than their EPE threshold, or an average amount of precipitation. The first two scenarios are of particular interest to test the hypothesis of a drier air mass reaching the other sites after precipitating at the first site. Table 4 summarizes the results for both datasets. In this table, "Above EPE thr" (i.e., above EPE threshold) refers to the percentage of cases where the other sites receive more precipitation than their site-specific EPE threshold, meaning that EPEs are simultaneously occurring at both sites, and "Neg. anom." (i.e., negative anomalies) refers to the percentage of cases with below-average precipitation.

| RACMO2.3 (%) | EPEs at IC | | EPEs at FK | | EPEs at TIR | |
|---|---|---|---|---|---|---|
| | FK | TIR | IC | TIR | IC | FK |
| Above EPE thr | 58.0 | 58.9 | 66.0 | 70.9 | 68.1 | 71.9 |
| Neg. anom. | 6.6 | 9.2 | 0.5 | 0 | 1.1 | 1.1 |
| Downscaling (%) | EPEs at IC | | EPEs at FK | | EPEs at TIR | |
| | FK | TIR | IC | TIR | IC | FK |
| Above EPE thr | 58 ± 4 | 57 ± 3 | 59 ± 4 | 66 ± 6 | 59 ± 3 | 66 ± 6 |
| Neg. anom. | 4 ± 2 | 5 ± 2 | 3 ± 1 | 1 ± 1 | 4 ± 1 | 3 ± 3 |

**Table 4: Distribution of the precipitation anomalies at two sites when there is a 95th percentile EPE at the third site, in %, for both the RACMO2.3 and downscaling datasets. For the downscaling, the average of the 10 members is shown, as well as the standard deviation to highlight the variability between members. See text for more explanations on the "Above EPE thr" and "Neg. anom." categories.**

For the RACMO2.3 results, EPEs at the IC site (Fig. 6a and Table 4) coincide less frequently to EPEs at the two other sites and more often result in negative precipitation anomalies (an average of 34 events) compared to the EPEs at FK and TIR (2–4 events on average). This therefore suggests that an EPE at IC may lead to drier conditions at the FK and TIR sites, possibly due to a reduced moisture availability in the air mass following intense precipitation at the IC site. Similar observations are made from the 98th percentile EPEs (Sect. S5 in the Supplement). For the EPEs at FK and at TIR, the distributions of the precipitation anomalies are very close, with EPEs occurring simultaneously in more than 66 % of the cases – compared to 58 % for EPEs at IC (Table 4). This indicates that precipitation variability at the FK and TIR sites is comparable according to RACMO2.3. However, longer distribution tails (e.g., beyond 20 mm) are observed for FK (Fig. 6a and 6e). This is particularly evident from the frequency distribution of precipitation anomalies during EPEs occurring at IC, where FK and TIR have similar percentages of events above their respective EPEs thresholds (Table 4) but the distribution tail is longer at FK (Fig. 6a). Overall, this indicates that FK is more subject to large precipitation anomalies than the other sites, a finding confirmed by the downscaling results (Fig. 6b and 6f).

For the downscaling results (Fig. 6b, 6d, and 6f), the internal variability of the ensemble, represented by the standard deviation of the ensemble, makes observing significant results more complicated, particularly for negative anomalies (see Table 4, lower rows). The EPEs at FK and at TIR indicate similar distributions for events larger than the threshold used for defining EPE, with a higher percentage of EPEs occurring simultaneously at TIR and FK (66 %) than at IC (59 %). EPEs at IC also coincide

less frequently with EPEs at the other sites (57-59 %). These observations largely align with the previous results from RACMO2.3.

In order to further investigate the spatial variability of precipitation and extreme precipitation events, scatter plots of normalized precipitation anomalies for pairs of sites are analyzed in Sect. S6 in the Supplement. This is performed for both the RACMO2.3 and the downscaling datasets and for both percentiles. The main results can be summarized as follows: (1) the unequal distribution of precipitation is more pronounced during EPEs; (2) EPEs induce a larger spatial variability than average conditions, primarily due to more localized effects; and (3) more extreme events result in more localized impacts. This applies to all three sites.

Statistical analysis was conducted on potential temporal trends for the simultaneous and non-simultaneous EPEs for the three sites in RACMO2.3 (not shown), and the results indicate an absence of significant temporal trend. These analyses were also conducted for the downscaling dataset (not shown), where we observed a few statistically significant trends for one or two ensembles for some sites; however, no general temporal trends could be identified.

Overall, the majority of the EPEs at one site coincide with EPEs at the two other sites (about 60 % of the cases), indicating that extreme precipitation events generally affect all sites, although more extreme events result in more localized impacts. In general, we cannot confirm the hypothesis of a drier air mass reaching the other sites after precipitating at the first site and instead point to the influence by comparable atmospheric conditions at the large scale, with similar air masses reaching the three sites. All these observations further support the previous results that neither precipitation nor EPEs explain the contrasting SMB trends in the three ice cores, at least based on RACMO2.3 and the downscaling dataset.

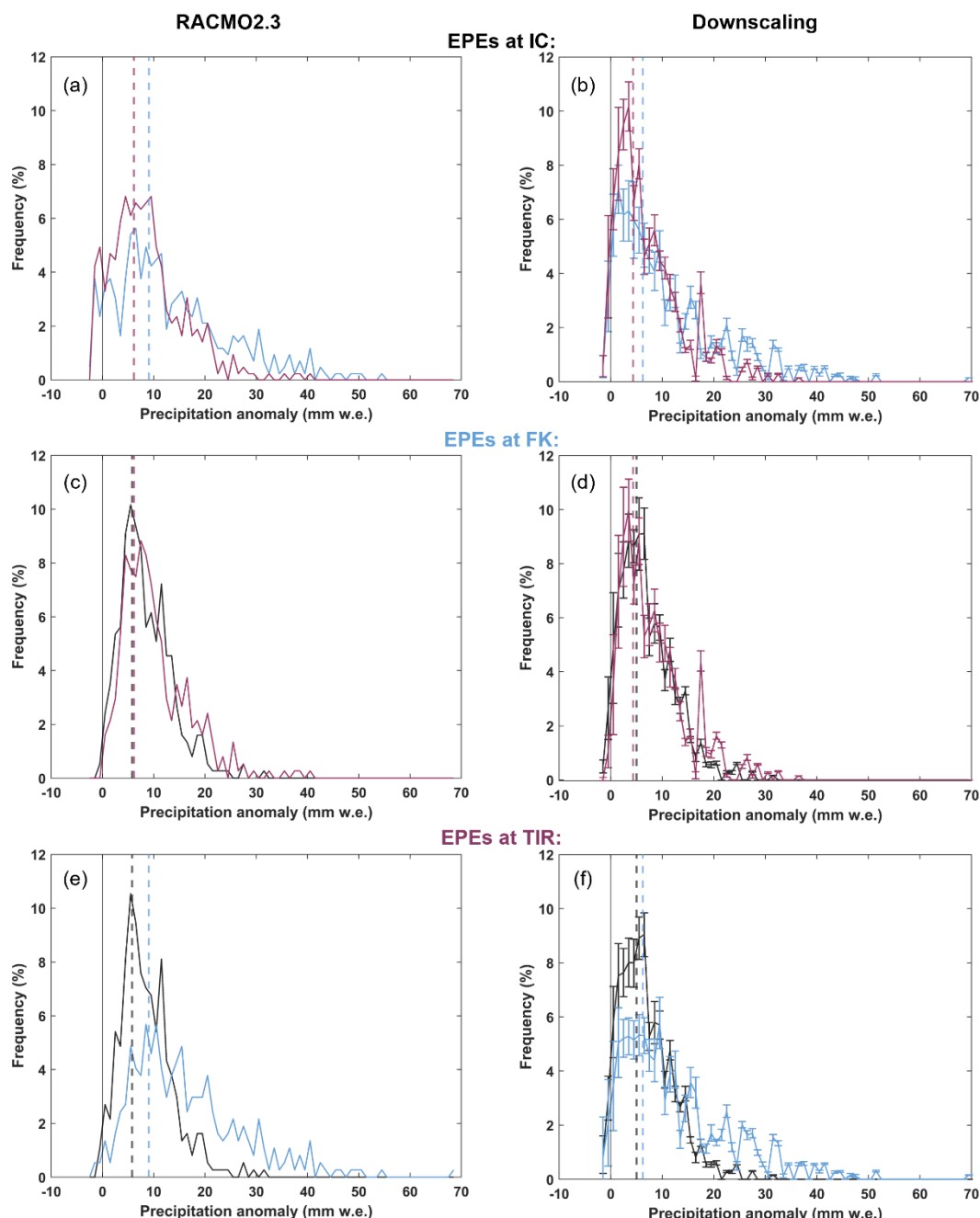

**Figure 6: Frequency distributions of precipitation anomalies (calculated as the precipitation − the average value of precipitation time-series) at the two other sites corresponding to the dates of 95$^{th}$ percentile EPEs at (a-b) IC, (c-d) FK, and (e-f) TIR, using the RACMO2.3 data series (left panels) and the downscaling data series (right panels). Colors of the distribution correspond to each site, with IC in black, FK in blue, and TIR in burgundy. The vertical dashed lines represent the EPE thresholds corrected for anomalies (EPE threshold − the average value of precipitation time-series).**

### 3.5.2 Synoptic patterns during EPEs at IC and negative anomalies at FK and TIR

Although the three sites experience very similar synoptic conditions during EPEs (Fig. 5), the 95$^{th}$ percentile EPEs at IC are more likely to result in negative precipitation anomalies at FK and TIR. Therefore, it is interesting to study the synoptic conditions during these events in order to potentially identify particular patterns (Fig. 7). Specifically, there are 28 and 39 cases of negative anomalies at FK and TIR, respectively, compared to only 0-4 negative anomaly events when EPEs occur at FK or TIR. Few of these cases happen with the 98$^{th}$ percentile EPEs (see Table S2 in the Supplement), they are not discussed further. From Fig. 7, these events appear to correspond to occasional, very localized situations where air masses precipitate at the IC site and arrive drier at FK and TIR sites. This contrasts with the strong and deep low-pressure system shown in Fig. 5a, which represents the atmospheric conditions during all EPEs at IC. The pressure gradient is considerably weaker with reduced wind strength in Fig. 7, indicating less effective moisture transport to FK and TIR.

**Figure 7: Maps of the sea level pressure anomaly (SLP) overlaid by mean surface wind vectors, retrieved from ERA5, during 95th percentile EPEs at the IC site leading to negative precipitation anomalies at (a) FK and (b) TIR. The number in parentheses corresponds to the number of cases. Blue colors indicate negative pressure anomaly, and brown colors indicate positive anomaly.**

## 4 Discussion

East Antarctica has recently emerged as the source of most uncertainties regarding Antarctica's potential to mitigate sea level rise (Eswaran et al., 2024). In the coastal DML region, ice-core records exhibit contrasting SMB trends. Multiple ice cores show a decreasing SMB trend over recent decades (Kaczmarska et al., 2004; Sinisalo et al., 2013; Schlosser et al., 2014; Altnau et al., 2015; Vega et al., 2016; Medley and Thomas, 2019; Ejaz et al., 2021), while one ice core indicates a significant increasing trend since the half of the 20th century (Philippe et al., 2016). Regional reconstruction by Thomas et al. (2017) suggests a significant positive SMB trend from 1800 to 2000. Recent observations by Wauthy et al. (2024) show one ice core with oscillating SMB and another with a marked decline. The three ice cores analyzed in this study, those of Philippe et al. (2016) and Wauthy et al. (2024), indeed exhibit highly contrasting SMB records, underscoring the strong regional and temporal variability of the SMB and its intricate nature. This complexity is, in part, attributable to the multitude of processes involved. As previously emphasized, a better understanding of the SMB is crucial to improve the projections of Antarctica's future contribution, and potential mitigation, to sea level rise.

Our aim in this study is to understand the variability observed in the SMB from three ice cores by analyzing the spatial and temporal variability of precipitation and extreme precipitation events using the RACMO2.3 and downscaling datasets. Similar observations are expected when comparing the downscaling data to RACMO2.3, as downscaling is derived from RACMO2.3. This is indeed the case, but some discrepancies are also noticed. While most precipitation quantity (> 95 %) falls during events affecting all three sites in both datasets, there are significantly more precipitation days in the downscaling results (85 % compared to 50 % in RACMO2.3). The average contribution and variance of the EPEs to the total annual precipitation are substantially higher in the downscaling data than in the RACMO2.3 data. There are also more EPEs per year in the downscaling dataset, particularly at the 95th percentile. These discrepancies may be indicative of a bias arising from the method employed, which focuses particularly on precipitation events in order to achieve statistical downscaling. Both RACMO2.3 and long-term downscaled simulations indicate no spatial variability in the distribution of precipitation days or temporal trends of precipitation across our three study sites, highlighting a significant model-data discrepancy with the SMB records from the ice cores. The frequency distributions of precipitation anomalies and the analysis of atmospheric conditions during EPEs suggest a common atmospheric pathway of air masses resulting in similar precipitation patterns at all three sites. This finding rejects the hypothesis of decreasing precipitation intensity along the air mass trajectory. Overall, the different temporal trends observed in the three SMB ice-core records cannot be attributed to precipitation-related processes based on these models.

Multiple studies emphasize the determining role of orography in shaping the EPEs occurrence and the precipitation distribution during these events (Turner et al., 2019; Gehring et al., 2022; Simon et al., 2024). The importance of finer scale surface topography is also underscored in the general precipitation pattern. For instance, it has been demonstrated that snowfall can be substantially higher on the windward side than on the leeward side of an ice rise (e.g., Lenaerts et al., 2014; Kausch et al., 2020). Moreover, Dattler et al. (2019) highlighted substantial spatial variability at the local, sub-grid, scale (< 25 km) related to wind-driven snow redistribution. This type of relationship between accumulation variability and surface topography could justify downscaling reanalysis accumulation product to 1 km grid in the future. Noël et al. (2023) presented a statistically downscaled version of RACMO2.3 SMB at 2 km resolution and compared it with the original 27 km resolution (1979-2021). While this downscaling relatively enhances the SMB at the continental scale (3 %) and reconciles modelled and satellite mass change, some significant differences between the two SMB products persist at the local scale. For example, the 2 km SMB product is approximately 300 mm w.e. yr$^{-1}$ higher than the 27 km product at the IC site. This is the order of the difference observed between the 5.5 km RACMO2.3 product and the IC12-derived SMB record (~265 mm w.e. yr$^{-1}$). This is consistent with the findings of Mottram et al. (2021), who reported underestimation of SMB in the low-elevation coastal regions of Antarctica by the 27 km RACMO2.3 product.

The tendency of models to underestimate SMB is evident when comparing the ice core-derived SMBs with those obtained from RACMO2.3 at 5.5 km resolution and from the downscaling dataset (Fig. 3). One hypothesis to explain the spatial model-data discrepancy could be related to missing processes, which are not related to the large-scale atmospheric circulation, in particular small-scale processes such as the underestimation of the blowing snow impacts on SMB in RACMO2 (Agosta et al., 2019). In a recent study, Gadde and van de Berg (2024) updated the blowing snow module in RACMO2.3 (using the coarse spatial configuration; 27 km) and reevaluated its contribution to the modelled SMB. Their findings indicate that the contribution of blowing snow sublimation increased by 52 % for the integrated AIS, corresponding to a SMB reduction, during the period 2000-2012 in the updated version of RACMO2.3 compared to the previous version. Concomitantly, a reduction of 1.2 % is observed in the integrated AIS SMB. Moreover, significant changes in blowing snow transport and total sublimation are observed in some coastal regions, including in the vicinity of our study sites. This suggests that the blowing snow term was underestimated in the RACMO2.3 version used in this study, yet also that there are high uncertainties associated with blowing snow-related processes. While wind speed can be particularly high at some ice rises (e.g., annual mean wind speed of 8–10 m s$^{-1}$ in the FK area; Simon et al., 2024), state-of-the-art atmospheric reanalyses struggle to correctly reproduce this low-level wind variability (Caton Harrison et al., 2022). Misrepresenting the effect of blowing snow, and other small-scale processes, could partly explain the different trends observed from the three ice-core records. Indeed, the limitation to accurately represent these small-scale processes is likely to contribute to the lower spatial and temporal variability observed in model outputs. The downscaling dataset of Ghilain et al. (2022) does not account for these sub-grid scale processes, as it exclusively represents snowfall. It should be noted that not all snowfall events result in accumulation. Souverijns et al. (2018) demonstrated that, during 38% of observed snowfall events at Princess Elisabeth Station in DML, freshly fallen snow is easily picked up by the wind and transported in shallow drifting snow layers throughout the event. They also observed that snowstorms of greater duration and spatial extent generally have a higher probability of resulting in accumulation on a local scale, while shorter events usually result in ablation. The downscaling dataset may also substantially underestimate low-level atmospheric circulation changes due to its reliance on RACMO2.3 outputs, which are based on the satellite period and exhibit a strong correlation between sites. This raises the possibility that past atmospheric circulation changes, which could account for the trends observed in the ice cores, are not accurately captured. The statistical downscaling indeed assumes a stationary relationship between precipitation and weather patterns, which may not be valid over longer timescales. The horizontal resolution of the models used also probably affects the results. Despite room for improvement, the fine spatial scale (5.5 km) of both the RACMO2.3 and downscaling datasets enables more realistic precipitation and EPE time-series to be captured. This

is particularly interesting in the context of EPEs, which have been shown to have a more localized impact than average precipitation, and in coastal regions where the topography is complex with variable and sometimes steep slopes. Furthermore, models with a higher resolution are expected to better represent atmospheric and surface dynamics. This highlights the importance of implementing small-scale processes and specific local topography in high-resolution models in the future.

Ice cores are also associated with uncertainties and biases. A single ice core record is affected by a multitude of processes that induce noise, non-climatic variability, and may even mask the true climatic variability. This non-climatic variability is called stratigraphic noise, which is introduced by irregular deposition, wind-driven erosion and the redistribution of snow (e.g., Hirsch et al., 2023). The effect of the stratigraphic noise has been particularly studied in the $\delta^{18}O$ profiles in snow and ice cores (e.g., Münch et al., 2016; Münch and Laepple, 2018; Casado et al., 2020; Hirsch et al., 2023). Altnau et al. (2015) studied 76 firn cores drilled in western DML and highlighted the noise present in both $\delta^{18}O$ and SMB records. The signal-to-noise variance ratio of SMB is about 0.4 in a single core from ice shelves, this drops to a value inferior to 0.1 for the firn cores from the plateau. Our three ice core sites are situated in local minimum SMB relative to their surrounding ice rise environments (Kausch et al., 2020; Cavitte et al., 2022). Therefore, the ice core SMB records are only representative of a limited area, approximately 200 to 500 meters in radius around the drilling site, and consistently exhibit lower SMB values compared to the mean across the entire ice rise which is estimated to be 70-150 mm w.e. yr$^{-1}$ higher (Cavitte et al., 2022). Consequently, the discrepancy between RACMO2.3 outputs and field observations would increase further if the ice core SMBs were adjusted towards higher values to reflect the consistently higher average SMB of the entire ice rise. Although local dynamic conditions at these drilling sites may influence the absolute accumulation values, ice-penetrating radar observations suggest that the ice-core records accurately represent the temporal variability at a multi-year resolution (Cavitte et al., 2022). However, the annual temporal variability, which may be affected by stratigraphic noise, appears to be underestimated in model outputs in comparison to the ice core observations, highlighting a potential limitation in the model representations of SMB interannual variability.

In summary, although both ice core observations and model outputs are subject to uncertainties and limitations, the contrasting temporal variability of the SMB records from the three ice cores cannot be attributed to differences in precipitation or EPEs patterns, according to state-of-the-art, polar-oriented regional climate models. Furthermore, it can be concluded that a common regional atmospheric circulation pattern, characterized by a dipole with low-pressure west and high pressure east of the ice rises, similarly influences all three sites. However, results of precipitation anomaly distributions indicate that, rarely, certain EPEs at the IC site occasionally lead to negative precipitation anomalies – indicative of drier conditions – at the FK and TIR sites, both connected to the grounded ice sheet. A key finding common to both datasets is that the EPEs have a more localized impact than average precipitation conditions. While this could explain some of the spatial variability observed in the three ice core records, it does not account for the different temporal trends, since no temporal trends in the EPEs were identified in either dataset.

Consequently, processes other than precipitation are probably driving these SMB contrasts. These could include multi-decadal variability, which may be underestimated in the dataset of Ghilain et al. (2022), local-scale processes such as blowing snow and erosion-deposition dynamics, or processes not included in atmospheric models, such as surface ice dynamics. Future research could focus on these processes, for example through the use of data assimilation methods that combine observational data (typically SMB from the ice cores) with the physics of climate models. This approach would allow for an evaluation of the ability of this method to reproduce the SMB variability observed in the ice-core records.

**5 Conclusions**

The AIS is a potential significant contributor to future global sea level rise, making it crucial to understand the mechanisms governing its SMB for accurate projections of future sea level changes. Our investigation into the SMB variability observed at three ice rises in coastal East Antarctica reveals complex interactions between atmospheric processes and local-scale dynamics. The analysis, based on a combination of regional climate modelling and statistical downscaling techniques, sheds light on the intricate nature of SMB variability. Previous studies have emphasized the importance of precipitation, including

its spatial and temporal distribution. While precipitation accounts for over 90 % of the SMB variance for each site according to RACMO2.3, our findings suggest that precipitation alone cannot fully account for the observed contrasts in SMB records among the three ice core sites.

Our findings, based on RACMO2.3 and downscaling datasets, indicate that synoptic-scale EPEs play a significant role in controlling interannual variability in precipitation and SMB. With contributions for the 95$^{th}$ percentile ranging from 30 to 50

545 %, depending on the dataset and site considered, these events contribute substantially to the total annual precipitation, underscoring their importance in shaping SMB patterns, particularly in coastal regions. Furthermore, evidence demonstrates that EPEs induce a larger spatial variability than average precipitation conditions. However, the absence of significant temporal trends in precipitation and EPEs across the study sites suggests that other processes may be driving the observed SMB contrasts.

Local-scale processes, such as blowing snow and erosion-deposition dynamics, may contribute to SMB spatial variability, as suggested by the discrepancies between ice core observations and model outputs. Additionally, the influence of surface ice dynamics, not fully captured by current models, cannot be discounted. Future research efforts should focus on developing high-resolution models that accurately represent topography and on integrating observational data, such as SMB records from ice cores, with advanced modelling techniques, including data assimilation methods. This approach has the potential to

improve our understanding of SMB dynamics and refining projections of Antarctic ice mass change and its implications for global sea level rise.

Our study underscores the complexity of Antarctic SMB variability and highlights the necessity for comprehensive, interdisciplinary approaches to elucidate the underlying processes driving these variations. Only through continued research can we enhance our ability to accurately predict the future evolution of the Antarctic Ice Sheet and its contribution to sea level

rise.

**Data availability**

The SMB and precipitation datasets from RACMO2.3 at a 5.5 km resolution are available from Willem Jan van de Berg upon request. The high-resolution precipitation downscaling from Ghilain et al. (2022), are available on Zenodo under Creative Commons Attribution 4.0 International Public License. The daily fields are separated into three parts:

https://doi.org/10.5281/ZENODO.6355455 for Part 1, https://doi.org/10.5281/ZENODO.6359385 for Part 2, and https://doi.org/10.5281/ZENODO.6362299 for Part 3. Sea-level pressure and wind fields from ERA5 can be retrieved at Copernicus Climate Change Service (C3S) Climate Data Store (https://cds.climate.copernicus.eu/datasets/reanalysis-era5-single-levels?tab=overview).

**Author contribution**

SW and QD designed the study. QD provided most of the raw data and made the SLP anomalies maps. SW performed the analysis and made the other figures. SW and QD both discussed the results and contributed equally to the writing of the manuscript.

**Competing interests**

The authors declare that they have no conflict of interest.

**Acknowledgements**

The authors would like to thank Hugues Goosse and Jean-Louis Tison for their insightful guidance and their critical reading of the manuscript, as well as Willem Jan van de Berg for sharing the RACMO2.3 outputs and Nicolas Ghilain for providing valuable advice on handling the downscaling data. The authors would also like to thank one anonymous reviewer and Aymeric Servettaz for their helpful and constructive comments, which have strongly improved the quality of this paper.

**Financial support**

During the course of the study, QD was a Research Fellow within the F.R.S.-F.N.R.S. (Belgium) and SW was supported by the Mass2Ant project (Belgian Science Policy Office - BELSPO, contract no. BR/165/A2/Mass2Ant) and the PARAMOUR project (under the Excellence of Science (EOS) program supported by the F.N.R.S. and the F.W.O. (Belgium); grant no. O0100718F, EOS ID no. 30454083).

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
