# Peer review of "Role of precipitation and extreme precipitation events on the variability of ice core surface mass balances in Dronning Maud Land: insights from RACMO2.3 and statistical downscaling"

_EGUsphere, 2025_

## Author Comment (AC1)

**Referee Comments 2 – Aymeric Servettaz**

This study makes use of carefully obtained datasets to discuss a recently important topic in Antarctic Climate science, the role of extreme events on driving the spatial and temporal variability of Surface Mass Balance.

The article structure is organized in a comprehensive way, with good progression. The manuscript reads well. Most conclusions are sound and supported by the data, but some affirmations remain to be contextualized, clarified, or sourced. The analysis and figures are of good quality. However, I think that the model data is not fully used, and even though principal results are already included in the article, there is room for further improvements suggested later in this comment. Finally, I am not competent enough to correct all, but I noticed some grammar issues that may result in inaccurate or even incorrect sentences.

In that regard, I support its future publication in The Cryosphere, but I think it would greatly benefit from major revisions before publishing.

*We thank the reviewer for taking the time to thoroughly review the manuscript and for providing constructive and detailed feedback. The reviewer's comments and suggestions have been addressed, please see the responses below.*

**1. Major comments and important revisions**

**Mistitled sections**

I picked up numerous cases of sections with titles that did not correspond to the content of the section. Sections are also unequal in length, so it would be possible to group ideas in sections of comparable length before deciding on an appropriate title.

*Regarding the unequal length of the sections, some will be merged (i.e., 3.2.1 and 3.2.2) and some will be reorganised, notably to directly address the differences between SMB observations from ice cores and models in the results (as asked by the Referee 1). Here is the suggested structure:*

*3 Results*
*3.1 SMB: comparison between the ice cores and the models*
*3.2. Contribution of the SMB components in RACMO2.3*
*3.3 Precipitation: spatial variability and temporal trends*
*3.4 Extreme Precipitation Events*
*3.5 Frequency distributions of precipitation anomalies*

**Title:** The article title emphasizes ice cores and does not mention RACMO, whereas the results section of the article has exactly 5 lines about ice core results (148-152) on a section of 192 lines (148-339). It is totally ok to study the components of SMB in a model in order to better understand ice cores composition, but the title should be a brief summary of what is in the study, not a summary of the final objective. In fact, it is deceiving to read "observed in three ice cores" the "role of (…) extreme precipitation events" when the actual results about EPEs rely on modelling.

*Thank you for raising this, we agree that the title should contain the notion of modelling. Suggestion: "Role of precipitation and extreme precipitation events on the variability of ice core surface mass balances in Dronning Maud Land: insights from RACMO and statistical downscaling". This version of the title emphasises the use of models.*

**L147:** May use a more descriptive title for the subsection

*Suggestion: Contribution of the SMB components in RACMO2.3*

**L178:** Section named interannual variability appears to treat spatial variability across the three sites, at daily resolution. The title is not suited.
**L200:** "Multi-decadal variability from downscaling". This section has two sentences related to the title, and the rest discusses spatial variability and annual variability.

*Thank you for pointing out these errors. We will merge those two sections (3.2.1 and 3.2.2) and only keep "3.2 Precipitation: spatial variability and temporal trends", to avoid unnecessary sub-division, also allowing to obtain more equal length between the sections.*

**L288:** distribution (of what?). Frequency distributions of precipitation anomalies would be a more informative section title

*Thank you for the suggestion, we will change the subsection title.*

**Introduction**

The first two paragraphs of introduction, about contextualization, are inaccurately describing or entirely missing core mechanisms.

*Although we believe that the first two paragraphs provide a general context to our study, we will revisit this section in light of the reviewer's comments and changes made throughout the manuscript.*

**L33:** "Basal mass loss" sounds like the only mass loss was to basal melting, whereas the general agreement is that glacier discharge is the main driver of ice loss (Davison et al., 2024; Diener et al., 2021). Basal melting can induce more ice loss, notably through reduction of buttressing effect and acceleration of ice discharge (Pritchard et al., 2012), but is not the main driver of Antarctic mass loss. (also mentioned **L37**)

*Indeed, it is not exact, we will remove the basal word on the two lines.*

**L46:** "Synoptic-scale dynamics correspond to short-lived intrusion of maritime air (…)": Worded like that, the synoptic scale dynamics always cause high precipitation, where in fact it depends on the scale and intensity of the synoptic system, and intense precipitation are localized. Smaller rates of precipitation are also observed at a distance from the center of the system, but still associated with it.

*We will rephrase this sentence and slightly modify the paragraph: "The two other mechanisms are related to dynamics, which can be further separated into large-scale dynamics and synoptic-scale dynamics (Dalaiden et al., 2020). Large-scale dynamics correspond to the southward moisture transport from lower latitudes (Marshall et al., 2017),*

*while short-lived intrusions of maritime air resulting in high precipitation are generally related to synoptic-scale dynamics. Such short-lived intrusions correspond to extreme precipitation events (EPEs) which can be associated with atmospheric rivers (Turner et al., 2019)."*

**L50:** "Large-scale dynamics correspond to the southward moisture transport from lower latitudes due to large-scale atmospheric modes of variability": There appears to be a confusion between large-scale circulation (average circulation) and large-scale variability (variability of the circulation). This specific sentence is unclear if not incorrect. Since this paragraph appears to describe the general mechanisms leading to precipitation, please detail variability in a different paragraph.

*Thank you for pointing that out. We will remove the last part of the sentence: "Large-scale dynamics correspond to the southward moisture transport from lower latitudes ."*

**L52-55:** I really do not like the separation of thermodynamic and dynamic, since we could classify "interaction of winds with topography" as adiabatic changes in the air parcel (as in foehn wind), which is fundamentally thermodynamics (this is also mentioned by the authors **L55**)

*We understand your point of view, we will remove the "Unlike thermodynamic changes" part of the sentence.*

The second half of the introduction is well written and introduces the problematic adequately.

**Lack of justification for the downscaling dataset**

The downscaling dataset provides useful insights onto the long-term evolution of precipitations. However, it is used throughout the article to create some statistics about precipitation events and spatial variability, along with RACMO2, and with almost no difference in most cases. Since it relies on RACMO2 to perform the downscaling, I would have appreciated a discussion on this aspect, to understand why it is relevant for sections discussing spatial variability, since its results do not differ from RACMO.

Do you want to verify if spatial variability is stable through time? In that case you may have to compare different epochs, using temporal subsets of the downscaled dataset.

Moreover, the contribution of EPEs to total SMB derived from the downscaled dataset are almost absent from the discussion, although I was expecting this to be one of the main results from this article after reading the introduction. This may also relate to the too low confidence in the downscaled dataset compared to ice core SMB, but I think both are worth considering (see the other hypothesis that I give for model-data discrepancy).

*The main interest of the downscaling dataset is that it relies on RACMO and covers a longer time period with a higher spatial resolution than coarseESM simulations. This allows us to potentially identify longer-term trends that cannot be observed in RACMO driven by atmospheric reanalysis over the satellite period and to study long-term spatial variability. We*

*attempted to make this interest clearer by developing the downscaling section (see one of the next comments).*

*We agree that the results of downscaling have not been used sufficiently. More attention will be paid to these results in the revised manuscript, we will look into how they compare to the RACMO results and what are the limitations. Regarding the model-data discrepancy - and the hypotheses to explain it -, it will also be clearly addressed earlier in the manuscript.*

Lines where the use of the downscaled dataset is not sufficiently justified:

**L138-145:** This section needs further description. Is RACMO2.3 run through the entire period ? or are daily climate states statistically linked on the observation period and then back-casted onto the longer simulation period? This doesn't need to go as much in detail as the cited article for downscaled dataset creation, but now please include enough detail so that we can understand the general method that was used to produce this dataset.

*We agree that the downscaling section could be more detailed. We will modify the text as: "Considering the strong internal variability of the Antarctic climate system (Jones et al., 2016), the satellite period might be too short to study the variability of the SMB. An opportunity to analyze longer periods rises from statistical downscaling which extends the time period covered by the RACMO simulation. Here, we use the dataset from Ghilain et al. (2022a) that employed a statistical downscaling method combining the daily snowfall from RCM of high spatial resolution to specific weather patterns observed in state-of-the-art atmospheric reanalyses. Based on these dynamical relationships, Ghilain et al. (2022a) applied this relationship on ensemble of simulations performed with a Earth System Model covering the 1850–2014 period to downscale precipitation over the coastal region of DML at high resolution (5.5km; see for details and specifications). Briefly, this dataset provides daily snowfall by using the RACMO2.3 simulation at 5.5 km horizontal resolution, ERA5 as the atmospheric reanalysis and an ensemble of 10 simulations performed with the Community Earth System Model version 2 (CESM2) at low horizontal resolution (1 degree; about 70 km). The resulting daily snowfall simulations combine the advantages of RCM and GCM since they are characterized by the high spatial resolution of RACMO2.3 and the longer record (1850-2014) of CESM2."*

**L142:** "high-resolution RACMO simulation" which simulation? the same as previously described?

*This information was lacking, we addressed this in our previous reply by detailing the downscaling technique.*

**L143:** is there a downscale simulation for each ESM simulation?

*Yes, each of the 10 members from the ESM is downscaled.*

**L196** "the results align with RACMO2.3": I was somehow expecting so since the downscaled dataset is a statistical tool using RACMO2.3
**L313:** "FK is more subject to large precipitation anomalies than the other sites, a finding confirmed by the downscaling results": same as above.

*It is indeed logical that RACMO2.3 and the dataset show similar results (even though there are some differences). We will state this clearly in the manuscript.*

**Affirmations not enough supported by the data presented**

**L198 – 199:** "confirming the absence of spatial variability in annual precipitation across the three sites"

This is an inaccurate shortcut:

Table 1 presents spatial distribution of event occurrence, and the distribution of precipitation amount during co-occurring events. You did not show the distribution of precipitation amount vs precipitation rate for each site, which could be very different from one site to another.

Simply said, you show here that it snows simultaneously (on about 80% of days) and that more than 95% of snowfall occurs on these 80% of days. In reality, one site could have very few variations in annual snowfall compared to the other two sites, that your statistics shown here would not capture, because they are daily and event-related statistics.

You should either reword your sentence to match what your data supports, or do the actual figure to support what you are writing. An example of a figure that you could possibly make is a distribution of annual precipitation for each site (relative to the mean accumulation), similar to your fig. 7 but not restricted to EPEs.

*We do show the annual precipitation series from RACMO2.3 for the three sites (Fig. S1) and point out that the correlation coefficients between the series are high and positive, which means that the temporal variability of precipitation at these three sites is highly similar, thus indicating a low spatial variability. However, we don't show them for the downscaled dataset that is the subject of the sentence. We have the equivalent data to Table 1 and Fig. S1 for the downscaled dataset and will use them in the revised version of the manuscript, to support the sentence. Regarding the distribution of precipitation amounts, please see our reply in the next comment.*

**L205-206:** "Overall, Table 2 confirms the absence of significant spatial variability in precipitation across the three sites."

Spatial variability is too vague in this case. I would prefer a "concurrent positive (or negative) annual SMB anomalies across the three sites", referring to the timing of anomalies, which is what your analysis reveals.

*We agree that the wording is too vague, thank you for the suggestion, we will use it.*

Again, same as Table 1, it would be possible and interesting to study the range of anomalies to answer the following question: Do stronger anomalies occur simultaneously at the three sites? (Not only showing positive or negative anomalies, but also comparing their values.) This could be simply presented by comparing sites by pairs, with scatter-plots of

accumulation anomalies. Plus, I think it might highlight that spatial variability at extreme precipitation rates is greater, as extreme precipitation is usually very localized. If this is the case it is an important result that would fit perfectly in your study.

*Reading this comment, we understand that we need to clarify the logic and the message of this manuscript. We will pay extra attention to this point when working on a revised version of the manuscript.*

*In answer to the question 'Do stronger anomalies occur simultaneously at the three sites?', this is what we are looking at by analysing the frequency distributions of precipitation anomalies (focusing on the EPEs at each of the three sites). However, we recognise that a more in-depth comparison of the spatial variability of precipitation and extreme precipitation events could yield interesting results (including developing the question of the effects of EPEs, as raised by the referee's comment at line 247).*

*To investigate this, we propose scatter plots per pair of sites with three categories: (1) all precipitation anomalies, (2) precipitation anomalies corresponding to the EPEs at the first site (and the corresponding values of the second site), and (3) precipitation anomalies corresponding to the EPEs at the second site (and the corresponding values of the first site). This means that certain values are found in two or more categories.*

*Normalized precipitation anomalies are expressed as a ratio of precipitation anomaly to the average precipitation day and are calculated as (precipitation - average precipitation day)/average precipitation day. Note that non-precipitation days (less than 0.02 mm per day) are assigned a value of 0 and therefore their ratio is -1. Dividing by the average of precipitation days removes the mean state (which is different for each site) from the variability and therefore makes it possible to compare sites without the influence of the amount of precipitation.Here we show an example for the 95th percentile and RACMO2.3.*

*An interesting feature to compare is the linear slope for each category. If the slope is equal to unity, this means that both sites receive the same relative quantities. In the Figure below, it appears that for most EPE cases - except for EPEs at the IC site - the slopes are lower than unity, indicating unequal distribution of the quantities. Another interesting feature is the explained variance ($R^2$). The explained variance is higher when considering all the events compared with the case where only the EPEs are selected. This therefore indicates that EPEs induce a larger spatial variability compared to the averaged conditions, mostly due to more localized effects.*

*Similar figures will be analyzed for both percentiles and datasets, shown and interpreted in the revised version of the manuscript.*

[Figure]

*Figure: Scatter plots of normalized precipitation anomalies per pair of sites - IC-FK (upper panels), IC-TIR (middle panels), and FK-TIR (lower panels) (a) all normalized precipitation anomalies, (b) EPEs at site of x-axis and corresponding anomalies at site of y-axis, (c) EPEs at site of y-axis and corresponding anomalies at site of x-axis, and (d) EPEs at both sites.*

**L222:** "percentile (98th) as a proxy for atmospheric rivers"

While it is true that atmospheric rivers can induce EPE, the definition of another percentile of EPE (98%) does not match the definition of atmospheric rivers given in the introduction, which needs to be a spatial feature with intense meridional moisture flux. If you want to link the 98[th] percentile with atmospheric river, you need further justification, either by citing literature proving (or showing here with other analysis) that 98th percentile of precipitation can be considered as resulting from atmospheric rivers in most cases.

The article by Wille et al. (2021) attributed *half* of EPE days within the 99th percentile to atmospheric rivers, which means that another half is not related to atmospheric rivers. This

proportion decreases for 95th and 90th percentile, which means that the 98th percentile that you used here should include atmospheric river-related EPE, but also include many other events not related to AR.

*We agree that considering the 98th percentile EPE as a proxy for atmospheric rivers is a shortcut. We will rephrase: "We use the 95th and the 98th percentiles. The 98th percentile corresponds to the largest 2% of daily precipitation events, including approximately half of the intense localized snowfall events produced by atmospheric rivers (Wille et al., 2021). ."*

**L292-293:** "to test the hypothesis of a drier air mass reaching the other sites after precipitating at the first site", **L303** "due to a reduced moisture availability in the air mass following intense precipitation at the IC site", and **L316** "hypothesis of a drier air mass reaching the other sites after precipitating at the first site"

Given that the three sites are coastal, a single air parcel precipitating at the three sites would require alongshore (zonal) transport, which goes against synoptic conditions presented in fig. 6. If you want to show that moisture is reduced after precipitation at IC site, you would have to show that the air parcel has gone through these sites exactly using air parcel tracking algorithm (e.g. hysplit or flexpart), or at least wind maps.

We thank the reviewer for this valuable comment. We agree that demonstrating reduced moisture availability along the air mass trajectory would ideally require air parcel tracking (e.g., HYSPLIT, FLEXPART) or detailed wind field analysis to confirm transport pathways. However, our intention was not to explicitly track the air parcel trajectory or quantify changes in moisture content along its path. Rather, we mentioned the possibility of a drier air mass downstream as a possible hypothesis to explain potential spatial variability in precipitation due to interactions between synoptic flow and local topography. Importantly, our analysis does not reveal significant spatial variability in precipitation across the three sites during the studied events. Therefore, while the hypothesis remains plausible in theory, we have not pursued a detailed moisture transport analysis, as the data do not show a compelling case for it.

Another explanation for the difference for IC site would be related to its particularities: IC is a dome summit on island (as opposed to coastal crests for the two others), therefore there are more "angles" from which the precipitation could form. This would be apparent not only on parcel tracking, but also on moisture flux maps, which have much thinner spatial variability than geopotential anomaly maps.

If you look closely, you can even see that effect on your Fig. 6, where the core of <-10 hPa of SLP anomalies is geographically more restricted for EPE at the IC site, meaning that the low-pressure system can be displaced and still produce EPE.

**L356-357:** "This finding excludes the hypothesis of decreasing precipitation intensity along the air mass trajectory" this sentence may need adjustments depending on the corrections you plan regarding my previous comment.

*Figure 6 shows the maps of mean sea-level anomaly, which is indicative of atmospheric circulation (i.e., the geostrophic air flow). This Figure suggests a rather zonal circulation along the coast. Simon et al. (2024) demonstrates a clear zonal circulation during EPEs using a map with vectors of mean wind (see their Fig. 2a), in accordance with our interpretation of the map. We will produce our own version of this map, using the dates of the EPEs defined with RACMO2.3 and the ERA5 wind fields.*

**L317-319** "All these observations further support the previous hypothesis that neither precipitations nor EPEs explain the contrasting SMB trends in the three ice cores and instead point to the influence of global atmospheric pathways"

First, the hypothesis you made earlier was the contrary: "A particularly high precipitation event at one site might result in abnormally low precipitation at the two other sites, possibly explaining the different SMB trends observed in the three ice cores" (**L279-280**). This does not further support it, this disconfirms it.

*The word "hypothesis" will be removed, as it is seems to refer to the hypothesis of a drier air mass after high precipitation at one site, while we meant that the results of frequency distributions support the previous \*observations\* (from the previous sections) that precipitation and EPEs do not explain the differences observed in the ice cores.*

Second, you still have 40% of EPEs that do not occur simultaneously, how could you rule out that they do not play a role in the differences observed between ice cores? You would have to show that there is no specific temporal pattern for the non-simultaneous EPEs.

*In lines 247-250, we indicate that there are no temporal trends in annual (simultaneous + non-simultaneous) EPE series, for both datasets. In this "Frequency distributions of precipitation anomalies" section, we do not analyze the temporal trends of the simultaneous EPEs. We did statistical testing on potential temporal trends for the simultaneous and non-simultaneous EPEs (95th percentile) for the three sites in RACMO and the results indicate no significant trend. We also did these analyses for the downscaled dataset, some members show significant trends for some EPEs (both simultaneous and non-simultaneous) for some sites, but no general trends could be identified. This will be specified in the new version of the manuscript.*

Third, what is "global atmospheric pathways" and how can the similarity between local EPEs explain less difference than global phenomena?

*We meant that the three sites seem to be influenced by comparable atmospheric conditions, with similar air masses reaching them. Moreover, the use of the "instead" word makes the sentence confusing. We will rework this sentence.*

**L326:** "global atmospheric pathways are predominantly observed"

I do not think I understand what you mean by this. Global atmospheric pathways would generally refer to atmospheric cells or jet streams, which are not presented in this study's result.

*We agree that this wording is imprecise, what we meant is that the synoptic conditions are globally very similar at the three sites during EPEs. We will change this sentence.*

**L386:** "the suitability of the 5.5 km products in this study."

Suitable in this study for which purpose? for site-to-site comparison? I think it needs a little more justification than just comparable correlation coefficients. The average values you described just before and on Fig. S4 seem quite different, so maybe the temporal variability of precipitation and precipitation intensities can vary quite much between the two spatial resolutions. In addition, if you have high annual SMB correlation, it is hard to say anything about EPEs which are daily events.

*As explained later in the comments, we do think that RACMO at 5.5 km resolution is the most suitable version for these analyses. This model configuration is based on the Antarctic configuration of RACMO2.3 – which has been extensively evaluated (van Wessem et al., 2018) – but adapted to the Dronning Maud Land region by enhancing the horizontal spatial resolution (5.5 km vs 27 km in the original Antarctic configuration). We will rewrite the justification in the revised version of the manuscript (especially since we will expand on the comparison between the different models) and detail the implications of using one version rather than the other.*

**L390-392:** "However, this temporal variability appears to be underestimated in model outputs in comparison to the ice core observations, highlighting a limitation in the model representation of SMB interannual variability"

Underestimation of temporal variability in models is one possible explanation that the authors explored well.

However, the temporal variability of ice cores could also be increased by non-climatic factors. Several studies have shown that: e.g. due to snow erosion-redeposition effects, creating uneven layers of accumulation (Karlöf et al., 2006; Münch et al., 2016; Münch and Laepple, 2018). A single ice core is usually overestimating the variability because of this effect. A study of intercomparison of 76 ice cores (Altnau et al., 2015), although inland DML, showed that signal to noise ratio for an ice core SMB is about 0.4 on shelves (meaning 60% of the observed variability is noise), or even less for continental sites (or even more noise). Similarly radar studies point to an overestimated SMB in single ice cores (Cavitte et al., 2023).

Given that the SMB is one of your datasets, you need to develop and present this second hypothesis in your discussion, to present the limits of this dataset.

*We agree that this hypothesis should be addressed. Thank you for the suggestion and references, we will carefully read them to expand on the discussion.*

**2. Minor comments and technical corrections**

**L21:** This sentence sounds like RACMO2.3 was further downscaled, which is not the case. Please rephrase.

*We can modify the text as: "Our results, based on RACMO2.3 and statistical downscaling-based dataset, confirm that precipitation is the primary driver of SMB".*

**L22:** please include numbers on the actual contribution of precipitation and EPE-related precipitation to SMB in the abstract

*With three sites and two different datasets, adding all the information requested will make the abstract less fluid. However, we will add this information in the conclusion, where the main results can be more detailed.*

**L35:** The Clausius-Clapeyron describes the moisture content in the air. The effect on precipitation is indirect, so this sentence should be reworked.

*We will remove the mention of the Clausius-Clapeyron law to avoid confusion.*

**L44:** I am unsure where this classification of "three mechanisms controlling precipitation" comes from.

*We agree that there is a reference missing (Dalaiden et al., 2020) and specify that they control precipitation variability.*

**L95:** Additional Information that could be included: Ice thickness at each ice rise, Synthetic Aperture Radar-based ice velocity (just to confirm that value at the crest is approximately zero; SAR ice velocity maps are not necessarily required, but could be cited)

*This type of data can be found in Wauthy et al. (2024). We added the elevation (above ASL) of the three ice rises, as it can be useful information for the atmospheric circulation.*

**L102:** "annual layer thickness" I guess you mean water-equivalent annual thickness? It needs to be specified here.

*We will add it to the text: "The water equivalent annual layer thicknesses are obtained from combining raw annual layer thicknesses with density profiles".*

**L103-104:** I understand that the correction is for ice flow divergence, thinning the layers over time, which is different from compression effects as would be inferred from the second half of this sentence. These two sentences could be reworked to use more specific wording: (1) Correction of compression effects using ice density profiles, and (2) Correction of thinning effects related to divergent ice flow at ice rises

*Indeed, vertical density changes and horizontal deformation/thinning of the (underlying) ice are two distinct effects but both are linked to the compression of snow under its own weight. We will rework the sentences to clarify them.*

**L111-117:** This paragraph may be more suited for results/discussion section, not methods?

*We understand this point of view, this paragraph will be transferred in the new result section comparing ice core observations and model outputs.*

**L126:** I had a hard time figuring out this "respectively". I am unsure about the grammar, but for me it would be more natural to place it at the end of the section (or at the beginning), not in the middle. Here I stopped to "respectively" and read as if SU is sublimation again, and ER is erosion deposition.

*We will rephrase the text as: "$SU_s$ represents the surface loss by sublimation and by evaporation, $SU_{ds}$ is the drifting snow sublimation and $ER_{ds}$ corresponds to the drifting snow erosion/deposition - drifting snow is caused by the near-surface winds (i.e., blowing snow) - and RU is the meltwater runoff." to avoid confusion.*

**L127:** I do not understand how the sublimation of drifted snow ($SU_s$) and erosion ($SU_{ds}$) are counted. If snow is eroded than sublimated, does it count in SUds? in the case that it is eroded but deposited elsewhere it does in ERds? but what is the difference from the viewpoint of ice core site, since it is just lost to erosion in the beginning, why not group them for simplification?

*As clarified in the previous comment, there are two terms for the drifting snow: sublimation ($SU_{ds}$) and erosion ($ER_{ds}$). We followed the definition given in e.g., Lenaerts et al. (2012) as it is the one used in the RACMO model. If drifting snow is first eroded, it counts as $ER_{ds}$, if first sublimated, then $SU_{ds}$. Erosion/deposition are both considered in the $ER_{ds}$ term, but the sign will change (- if deposited, to get a positive contribution from the equation, opposite for erosion). From the viewpoint of an ice core, it is not important as only the final result (SMB) is obtained, however, we thought it was important to highlight how the SMB is built in RACMO since we use these data throughout the paper.*

**L148:** RACMO is not black on Fig. 3, and there is no grey line for ice core records.

*Thank you for pointing that out, RACMO is indeed in dark blue and the ice core records are in black.*

**L150:** FK, TIR, and IC were previously referred to using Fk17, TIR18 and IC12. May need to homogenize notations.

*FK17, TIR18 and IC12 are the names of the ice cores, while FK, TIR, and IC refer to the data of the grid cells corresponding to the ice core locations. Here we use the more generic term (i.e. FK, TIR, and IC).*

**L203:** "in contrast with ice core records" please ref fig. 2 here, or include the ice core records on the Fig. 4

*We will add the reference to Fig. 2 in the text.*

**L217:** I would prefer to rephrase this sentence to avoid starting with "Besides". This would be acceptable if "besides" refers to the previous paragraph, but this is not the case here.

*We will change "Besides" for "In addition to".*

**L220:** you could merge the two sentences by replacing "a certain percentile" by "the 95th percentile". Please also precise if this is the percentile for the corresponding model grid cell.

*We will adapt the text to mention both percentiles and the corresponding model grid cell.*

**L240-251:** please move the cross-site correlation comparison after the description of results from Table 3. Discussion about the values (**L252-262**) should come before cross-site correlations.

*We will do so.*

**L238 (Fig. 5 caption):** "variability" I would rather use the term uncertainty to describe the internal variability of downscaling products, to differentiate from temporal or spatial variability.

*We mean internal variability due to the 10 members differing by their initial states. This is thus different from uncertainty, but we will add the "internal" notion to avoid confusion with temporal or spatial variability.*

**L247:** "EPE impacts are more localized compared to the average conditions". This is an important result from this study. I think it deserves to be discussed more, in light of the maps you made, and also be repeated in Conclusions.

*Thank you for pointing this out. We will indeed emphasize this important result in the revised version of the manuscript (including a comparison of the spatial variability of the precipitation and of the extreme events, see previous comment L205-206).*

**L270:** (Suggestion) It would be interesting to have also meridian moisture flux maps, that would enable discussion about atmospheric rivers. Such maps would be more relevant if shown on a smaller domain, similar to fig.1 for example. In that case, comparison of 98th and 95th percentile may highlight spatial differences, explaining the lower correlations for 98th percentile precipitation across sites given line 244. Furthermore, cross-site differences may appear more clearly at a finer scale.

*We will consider this option if our timing allows it. Thank you for the suggestion.*

**L307:** please explain what you mean by "evolve in the same direction": Geographical direction?

*This is a misuse of the expression. We will replace it by "is comparable": "the precipitation variability at the FK and TIR sites is comparable according to RACMO2.3".*

**L311:** As the figure title says, this is frequency distribution "of precipitation" not "of EPEs".

*Indeed, we will correct the text to make our point clearer.*

**L323:** "Colors of the distribution correspond to each ice core" not to each ice core (this is not ice core data) but to the model data corresponding to each ice core site.

*Thank you for pointing this out, we will change the text.*

**L350** "a better understanding of the SMB is crucial to improve the projections" It would be good to precise here how this work can contribute to better understand SMB. Importance of extreme events is one aspect. The strength of your model is also its small scale capable of capturing more realistic SMB (as in fig.8 of Ghilain et al., 2022), and this may be mentioned at some point. In particular, *how high spatial resolution affects the representation of EPEs and their contribution* could be of particular importance on coastal location with important terrain slope. You open this topic **L363** but do not develop further or discuss your data, only citing literature.

*Thank you for the suggestion of discussion that we will certainly use in the revised version of the manuscript. By modifying the structure of the results, we will try to clarify the logic followed in order to contribute to the understanding of the SMB and will recall/summarize it in this part of the discussion.*

**L352:** "simulations indicate no spatial variability" please precise that you are talking about trends; the average value for each site is different, which is a form of spatial variability.

*Indeed, we will precise this.*

**L364:** "snow redistribution" I was expecting an example of precipitation pattern, and I got snow redistribution. These two processes that admittedly depend on topography, but since you study the different components of the SMB, I would have liked a clearer separation between them.

*We agree that a clear separation should be made between the two processes, we will rework the text to ensure this.*

**L365:** "could enable downscaling reanalysis" consider replacing "enable" by "justify"?

*We will replace it.*

**L371**: "difference observed between the 5.5 km RACMO2.3 product and the IC12-derived SMB record (~265 mm w.e. yr$^{-1}$)." Does this mean 5.5km RACMO has still a large margin for improvement? Please comment on your model here.

*We will comment on the model in the new section dedicated to the comparison of the SMB records. To respond briefly, yes, there is still a margin for improvements in RACMO 5.5 km, as shown by Kausch et al. (2020) (see also the next comment).*

**L376:** How about comparison of SMB in the 2 km scale RACMO and the ice cores? Are the SMB getting more similar to ice core data in the 2 km product?

*Regarding the comparison with the ice core SMBs, the 2 km scale RACMO is closer than the one of the 5.5 km scale for IC and TIR, but not for FK. However, as the downscaling dataset is based on the 5.5 km scale, we made the choice to use this version throughout the paper,*

*to analyse "comparable items". Moreover, the 5.5 km scale RACMO has been well evaluated in our region of interest (i.e., Lenaerts et al., 2014; Kausch et al., 2020; Ghilain et al., 2022) and we had easy access to the daily datasets because of previous projects we were involved in while the 2 km RACMO dataset has been only publicly archived at annual resolution. In addition, the 2 km RACMO results from the statistical downscaling of the 27 km scale RACMO with a high resolution surface topography at 2 km. RACMO 5.5 is thus expected to better represent the atmospheric and surface dynamics.The comparison we made between the 5.5 and 2 km scales aimed to recognize that the resolution used to analyze RACMO has a potential impact on the results. However, we realize that mentioning the 2 km RACMO dataset adds confusion in the manuscript. Instead of conducting the comparison in the discussion, we will propose the use of models with a finer resolution (both for RACMO and downscaling) as well as enhanced representation of processes, in particular processes related to the blowing snow, as a perspective to develop in the future.*

**L378:** "the three SMB time series" please precise "annual SMB" if this is the case.

*It is indeed the case, we will add it in the text.*

**L380:** you can recall that FK and TIR are the two sites connected to the continent with grounded ice sheet (here, but also at other points of discussion)

*It is indeed interesting to recall that FK and TIR are ice promontories connected to the grounded ice sheet.*

**L388:** "would increase further if the ice core SMBs were adjusted to reflect the consistently lower SMB" Are you considering to adjust ice core SMBs? I do not understand this phrase. It would be possible to change corrections made to ice cores to produce accumulation record, but it needs more contextual discussion.

*We are not planning to adjust the ice core SMBs, we wanted to raise the point that it was a possibility if there is sufficient radar data to do so, but we understand this can be confusing, and will elaborate on the difference between the local SMB record from the ice core and the ice core representativeness of the surrounding ice rise.*

**L390:** "the temporal variability is well represented by ice-core records (Cavitte et al., 2022)." It might be good to remind here what are ice-core records compared to in Cavitte et al. 2022, to confirm that temporal variability is "well represented"

*Thank you for mentioning this, we will indeed add the information that this affirmation comes from ice-penetrating radar observations.*

**L397:** "contribution of blowing snow sublimation increased by 52 %" it may be good to precise that this means reduced SMB for Antarctica.

*This can be inferred from the next sentence but we can elaborate on this.*

**L407:** "the contrasting SMB records" again you need to precise <temporal variability of SMB>, because the relative contribution of EPEs to average SMB (fig.5) follows roughly the same distribution as average ice core total SMB.

*We will change the text for "the contrasting temporal variability of the SMB records".*

**L408-409:** "a common regional atmospheric circulation pattern" may be replaced by more detailed description such as "a dipole of Low-pressure West/High Pressure East of the ice core site"

*Thank you for the suggestion, we will replace the text accordingly.*

**L414-415:** "only representative of a limited area, approximately 200 to 500 meters in radius around the drilling site, and consistently exhibit lower SMB values than the mean across the entire ice rise" This affirmation can be easily sourced to previous works in this area, please cite them here

*It should indeed cite Cavitte et al. (2022), this reference will be added.*

**L430-451** The Conclusions are OK, but may need to be reworked to match the corrections made, emphasize more on current study's results, and give numbers, especially about EPEs: how much do they contribute to mean SMB, to SMB interannual variability, what atmospheric pattern lead to EPEs… This should answer the original question of "*are EPEs driving the variability at these sites?*". Maybe you can also point out that the EPEs impact is currently underrepresented at Antarctic scale, due to the technical aspect of fine scale modelling and coastal topography, which is one of the challenges that could actually be tackled in the short term, by doing circumpolar coastal modelling. There could also be conclusions regarding the potential of SMB to compensate for ice loss in the future, given that models seem to underestimate coastal accumulation based on your results.

*Thank you for the detailed review and suggestions. We will follow them and will adapt the conclusions to the corrections and changes made in the manuscript.*

**Supplement, Fig. S4:** how do you explain the relative difference lower than -100% for IC? Wouldn't 100% reduction just become zero?

*The relative difference (%) is defined as: $(v - v_{ref})/v_{ref} * 100$ with v the new value (here the SMB from RACMO at 2 km resolution) and $v_{ref}$ the reference value (here the SMB from RACMO at 5.5 km resolution).*

*This means that if the difference between the reference and the new value is higher than two times the initial value, we will obtain a negative value under 100 %. For example, in 1994, $v_{ref}$ = 252.6 mm w.e. $yr^{-1}$ and v = 553.6 mm w.e. $yr^{-1}$ and the relative difference is thus –119 %. However, we do understand that this might be confusing and we will change the Figure to express it in absolute difference $(v - v_{ref})$.*

**References**

Altnau, S., Schlosser, E., Isaksson, E., and Divine, D.: Climatic signals from 76 shallow firn cores in Dronning Maud Land, East Antarctica, The Cryosphere, 9, 925–944, https://doi.org/10.5194/tc-9-925-2015, 2015.

Cavitte, M. G. P., Goosse, H., Matsuoka, K., Wauthy, S., Goel, V., Dey, R., Pratap, B., Van Liefferinge, B., Meloth, T., and Tison, J.-L.: Investigating the spatial representativeness of East Antarctic ice cores: a comparison of ice core and radar-derived surface mass balance over coastal ice rises and Dome Fuji, The Cryosphere, 17, 4779–4795, https://doi.org/10.5194/tc-17-4779-2023, 2023.

Davison, B. J., Hogg, A. E., Moffat, C., Meredith, M. P., and Wallis, B. J.: Widespread increase in discharge from west Antarctic Peninsula glaciers since 2018, The Cryosphere, 18, 3237–3251, https://doi.org/10.5194/tc-18-3237-2024, 2024.

Diener, T., Sasgen, I., Agosta, C., Fürst, J. J., Braun, M. H., Konrad, H., and Fettweis, X.: Acceleration of Dynamic Ice Loss in Antarctica From Satellite Gravimetry, Front. Earth Sci., 9, https://doi.org/10.3389/feart.2021.741789, 2021.

Ghilain, N., Vannitsem, S., Dalaiden, Q., Goosse, H., De Cruz, L., and Wei, W.: Large ensemble of downscaled historical daily snowfall from an earth system model to 5.5 km resolution over Dronning Maud Land, Antarctica, Earth System Science Data, 14, 1901–1916, https://doi.org/10.5194/essd-14-1901-2022, 2022.

Karlöf, L., Winebrenner, D. P., and Percival, D. B.: How representative is a time series derived from a firn core? A study at a low-accumulation site on the Antarctic plateau, Journal of Geophysical Research: Earth Surface, 111, https://doi.org/10.1029/2006JF000552, 2006.

Münch, T. and Laepple, T.: What climate signal is contained in decadal- to centennial-scale isotope variations from Antarctic ice cores?, Clim. Past, 14, 2053–2070, https://doi.org/10.5194/cp-14-2053-2018, 2018.

Münch, T., Kipfstuhl, S., Freitag, J., Meyer, H., and Laepple, T.: Regional climate signal vs. local noise: a two-dimensional view of water isotopes in Antarctic firn at Kohnen Station, Dronning Maud Land, Climate of the Past, 12, 1565–1581, https://doi.org/10.5194/cp-12-1565-2016, 2016.

Pritchard, H. D., Ligtenberg, S. R. M., Fricker, H. A., Vaughan, D. G., van den Broeke, M. R., and Padman, L.: Antarctic ice-sheet loss driven by basal melting of ice shelves, Nature, 484, 502–505, https://doi.org/10.1038/nature10968, 2012.

Wille, J. D., Favier, V., Gorodetskaya, I. V., Agosta, C., Kittel, C., Beeman, J. C., Jourdain, N. C., Lenaerts, J. T. M., and Codron, F.: Antarctic Atmospheric River Climatology and Precipitation Impacts, Journal of Geophysical Research: Atmospheres, 126, e2020JD033788, https://doi.org/10.1029/2020JD033788, 2021.

---

## Author Comment (AC2)

**Referee Comments 1**

The authors have used the RACMO2.3 model and a downscaled, long-term dataset to investigate temporal and spatial differences in precipitation and extreme precipitation events (EPEs), and their role on Surface Mass Balance (SMB) at three different ice core sites in Dronning Maud Land, Antarctica.

The paper is well structured and flows logically from each section. The introduction and motivation outline the gap in our knowledge and the importance of investigating precipitation and extreme events. The discussion is also comprehensive and provides a different perspective on the results. However, moving some of the discussion points from the discussion to the results or introduction would provide the reader more trust in the results and the use of the model for assessing the precipitation. In addition, the results could be separated into spatial and temporal distribution to better aid the understanding. I do think major revision is required, as some of the results are not clearly explained or presented. I am also unsure about the use of the multiple datasets and how they compare or validate the observations and each other. I believe the study sheds light on an important topic, and showcases the difficulties of using relatively lower-resolution models to investigate SMB at site-specific observations, especially in a geographically complex region. I think the conclusions are valid and important, but the results need more clarification and justification before it can be published.

*We thank the reviewer for taking the time to review the manuscript and for providing constructive and precise feedback. The reviewer's comments and suggestions have been addressed, please see the responses below.*

Comments:

Your introduction is very comprehensive, and your motivation is clearly outlined. No comments for the introduction.

Major revision:

Section 2.3: *(1)* More information is required for this product. Add the time period to line 140 - 'Extending the time period covered by RACMO' is not enough for the reader to assess the length of time. Even though you later say the ESM was used from 1850-2014, it is not clear if the downscaled final product also uses this same timeframe. Which version of high-resolution RACMO is used? 2km or 5.5km? What resolution is the downscaled dataset? The reader shouldn't have to read Ghilain et al. 2022 for this information, given that it is important to the study.
 *(2)* The downscaled product uses RACMO data, but there are large differences in the outcome, especially for the EPEs, but this isn't reflected in results or discussion – you should discuss this.

*(1) We agree that the downscaling section could be more detailed. We will modify the manuscript to provide a more detailed description of the dataset: "Considering the strong internal variability of the Antarctic climate system (Jones et al., 2016), the satellite period might be too short to study the variability of the SMB. An opportunity to analyze longer*

*periods rises from statistical downscaling which extends the time period covered by the RACMO simulation. Here, we use the dataset from Ghilain et al. (2022a) that employed a statistical downscaling method combining the daily snowfall from RCM of high spatial resolution to specific weather patterns observed in state-of-the-art atmospheric reanalyses. Based on these dynamical relationships, Ghilain et al. (2022a) applied this relationship on ensemble of simulations performed with a Earth System Model covering the 1850–2014 period to downscale precipitation over the coastal region of DML at high resolution (5.5km; see for details and specifications). Briefly, this dataset provides daily snowfall by using the RACMO2.3 simulation at 5.5 km horizontal resolution, ERA5 as the atmospheric reanalysis and an ensemble of 10 simulations performed with the Community Earth System Model version 2 (CESM2) at low horizontal resolution (1 degree; about 70 km). The resulting daily snowfall simulations combine the advantages of RCM and GCM since they are characterized by the high spatial resolution of RACMO2.3 and the longer record (1850-2014) of CESM2."*

*(2) Thank you for highlighting this missing point. As detailed in one of the following comments, we will reformat the manuscript to better analyze the differences between the datasets (from models but also the ice core records) and focus more on the downscaling outputs in the results and discussion. We will also address the potential limitations of the downscaled dataset.*

Which variables are you using? Snowfall or precipitation? Are these synonymous or comparable between the downscaled dataset and RACMO? Does total precipitation = snowfall in this region, or are their times of rainfall in summer?

*We use precipitation (snowfall + rain) from RACMO2.3 and snowfall from the downscaled dataset. Differences between precipitation and snowfall are negligible - of the order of 0.1 mm yr$^{-1}$ (0.14, 0.13, and 0.07 mm yr$^{-1}$ for IC, FK, and TIR, respectively, while the average precipitation per year are 431.3, 497.0, and 360.19 mm yr$^{-1}$) in RACMO2.3. To make it simpler and avoid confusion, we will mention this negligible difference and will then use a generic term (precipitation) throughout the manuscript.*

Section 3: There are major differences between the simulated SMB by RACMO and the ice core records. This is the first thing reported and then makes it very difficult for the reader to trust that RACMO is going to be used for the rest of the study. The justification for using it comes in the discussion, but you should consider moving this earlier, and perhaps bolstering this justification further. Almost 50% of the ice core SMB is not represented in the model – if precipitation is the main component, are you convinced that the model is representing the precipitation properly? Whilst models are always wrong, there is additional model justification and testing which is presented in the discussion which could perhaps come early in the results to bolster the reason for continuing to use RACMO despite the consistent, large underestimation. It would be ideal to see more comparison of the key variables such as precipitation with other observations.

The discussion section regarding complexity of SMB in ice cores should perhaps be moved to the introduction or results. In my case, I am very familiar with RACMO and the SMB analysis in the polar regions, but haven't used ice cores as observations before. Therefore, a straight comparison of the ice core and the model seems like a bad idea, given how poorly

RACMO captures the ice core observations. However, I do see value in continuing the study to analyse the RACMO data and assume that it can be a tool for showing SMB differences in time/space.

*We understand that the differences between the ice core records and the model's outputs can be confusing and lower confidence in the results. This should indeed be addressed earlier in the manuscript. We will start the Results with a section dedicated to the comparison of the SMB from the ice core observations and from the models. From this point, we will reformat the manuscript to better highlight the logic we have followed. Here is a proposition of structure:*

*3 Results*

*3.1 SMB: comparison between the ice cores and the models*

*→ this section will show the time-series of precipitation from RACMO2.3 and the downscaled dataset, as well as SMB from ice core records (including adapted Figures) and explain the possible reasons and uncertainties behind the differences (both for the ice core observations and for the models). Note that we don't have the SMB data for the downscaling results (only the snowfall), this will be addressed.*

*3.2. Contribution of the SMB components in RACMO2.3*

*→ this section will show that, according to RACMO2.3, precipitation is the main driver of SMB. By relying on RACMO2.3, we can thus make the hypothesis that the SMB variability observed in ice cores is explained by the precipitation. RACMO, recognized as an excellent tool for modelling precipitation in polar regions, will therefore be used to study/evaluate the spatial variability of precipitation over Dronning Maud Land, in order to shed light on the large spatial variability observed from ice core records. In this section, we thus make the hypothesis that RACMO2.3 performs well enough to investigate the potential processes explaining the large spatial variability of precipitation. However, since RACMO spans a short period of time, we might under-sample the impact of internal variability on precipitation over Dronning Maud land. Therefore, we need to obtain longer time-series. We introduce the downscaled dataset from Ghilain et al. (2022).*

*3.3 Precipitation: spatial variability and temporal trends*

*→ here, we will merge the two sub-sections (3.2.1 and 3.2.2) and will rework the text to better show the main results (distribution of the precipitation days, trends, and correlation between the time series), including those of the downscaled dataset.*

*3.4 Extreme Precipitation Events*

*→ in this section, we will detail how we calculated EPE thresholds for both RACMO and downscaling datasets. More emphasis will be given on the downscaling results, and how they differ from RACMO outputs, as this downscaled dataset can be used to "increase the sample size" of EPEs. Potential bias towards higher precipitation of the downscaling technique, potential lack of spatial variability(see the reply to next comment) as well as*

*issues in the downscaling method of Ghilain et al. (2022a) will also be mentioned. Synoptic patterns during EPEs will remain in this section, with details on the use of ERA5.*

*3.5 Frequency distributions of precipitation anomalies*

*→ we understand this section needs to be clarified. More explanations will be given on the method and purpose of looking at the frequency distributions.*

*The discussion and conclusion will be adapted accordingly.*

With the downscaled data, it isn't even capturing the long-term trends found in ice cores (section 3.2.2), which makes it even more challenging to justify using. If it isn't capturing long-term trends, which typically models can capture, how do you know it is capturing any spatial variability?

*Ghilain et al. (2022) observed a general good agreement between the downscaled snowfall (precipitation) and eight ice cores located in the DML coastal region. This will be mentioned, as well as the clear indication that the downscaled dataset is not capturing the long-term trends of our records (in the section comparing SMB from ice cores and models).*

*Looking at the differences between EPEs from RACMO and the downscaling, we made additional calculations to obtain the average number of EPEs per year in both datasets and it appears that there is no clear spatial variability (in terms of number of events per year), as shown on the following table:*

|  | Downscaling (1850-2014) | | | RACMO (1979-2016) | | |
|---|---|---|---|---|---|---|
| EPE 95 | IC | FK | TIR | IC | FK | TIR |
| # EPEs | 2656 | 2599 | 2590 | 426 | 374 | 370 |
| avg # EPEs/year | 16.09 | 15.75 | 15.70 | 11.21 | 9.84 | 9.74 |
| EPE 98 | | | | | | |
| # EPEs | 1062 | 1040 | 1036 | 171 | 149 | 148 |
| avg # EPEs/year | 6.44 | 6.30 | 6.28 | 4.50 | 3.92 | 3.89 |

*This table shows that the downscaled product might be biased towards higher precipitations, thus overexpressing those, because of the way the downscaling is built - i.e., by focusing on the precipitation events. However, this does not mean that downscaling is significantly less effective than RACMO at capturing spatial variability. These important observations will be added to the manuscript.*

*Ghilain et al. (2022) highlight an important bias reduction in comparison to CESM2 without downscaling. This shows that, despite the existing room for improvements, the downscaled dataset is the best tool available to increase the sample size and extend the record.*

The authors give a short analysis of the blowing snow contribution – which is significant, but then it is not investigated further or mentioned in relation to the spatial differences in SMB between the three sites. Perhaps more emphasis should be given to this investigation, especially as you conclude that precipitation/extreme precipitation is not responsible for the spatial differences between the sites. If there is a blowing snow modified version of RACMO, could you investigate (briefly) the differences in the model output with and without this modification?

*We agree that the blowing snow contribution is a solid hypothesis to explain the models-observations discrepancies. It is presented along other hypotheses (small-scale processes, influence of the ice rise dynamics), which will be further detailed in the revised version of the manuscript. We would prefer to keep the focus of this paper on the precipitation/EPE and present the hypotheses not linked to precipitation as perspective for future work.*

Figure 3: It would be useful to change this to better allow the reader to compare the products and the locations. Figure 3 is busy and it is too hard to compare SMB from RACMO and observations, this could be a separate figure to the other components. In addition, it is hard to compare the locations when there's a lot happening in one figure. Similarly, it would be useful to get one figure where ice cores, RACMO and downscaled data are presented together. Statements like 'in contract with ice core records' (Line 203) are hard to check in the figures, when long-term SMB from ice cores is on figure 2, satellite-era RACMO is on figure 3 and long-term downscaled data is on figure 4.

*As detailed in a previous comment, the structure of the Results will be modified and the Figure 3 will be changed to a figure with the SMB from observations and models and a figure presenting the SMB components in RACMO. We will keep your suggestions in mind when doing the new figures.*

Section 3.3: More information is needed on how you calculate the thresholds – does each location have its own 95% threshold – e.g each grid cell which represents the ice core location has a value? Or are you using an area average for all locations? Are the thresholds re-calculated for the downscaled data? Later on, you use ERA5 too – are you calculating the thresholds again for ERA5 data, or simply using ERA5 to extra data on specific dates, which have been above the threshold from RACMO?

*Thank you for raising these questions, we understand that we need to elaborate more on these, we will give more details on how each EPE threshold is calculated. Briefly, each location and each dataset has its own threshold calculation. Regarding using ERA5, we use it as an additional data input, keeping the threshold from RACMO. This is discussed more in the next comment.*

Section 3.3.2: I think it is a good idea to look at the synoptic situation during these events, but I question the addition of ERA5 data (also because it is not listed in your data section). This is another dataset, which is not compared to RACMO, the downscaled data or ice cores. Whilst synoptic conditions are generally well captured in ERA5 and most models, you are looking at specific dates of these events. How do you know that ERA5 also captures the EPEs which RACMO is seeing? Precipitation is a difficult variable, even in higher resolution reanalysis products. Perhaps it is better to look at the synoptic conditions in RACMO, rather

than introduce an additional dataset and therefore additional uncertainties in the conclusions. If you do stick with ERA5 – there should be some discussion of its useability in this region and how it compares to RACMO. In addition, do you select the data from ERA5 based on dates in RACMO, or do you re-calculate the 95% threshold with ERA5 data?

*Thank you for identifying the missing information regarding ERA5 data in the data section, we will add it.*
*We indeed use ERA5 as an additional dataset allowing us to investigate the synoptic conditions during EPEs because the domain of RACMO2.3 used in the study is too small. We assume that ERA5 capturessimilar EPEs as RACMO, since RACMO uses the previous version of the ECMWF reanalysis (ERA-Interim) for the boundary conditions and the domain is not very large, so the atmospheric dynamics should be relatively similar. We will add this to the description of how the synoptic patterns are obtained.*

Section 3.4: The geographic and synoptic set up doesn't particularly align with the characteristics of foehn winds. With the Peninsula, a long, high ridge prevents the air from flowing around the obstacle and therefore forces it over 2000m– this is what creates the foehn winds. However, in the case of ice rises, the airflow could flow around the obstacles, given their size, and likely not create the warm and dry leeside conditions. The definition of a foehn wind is also the warm and dry lee slope winds, and not the reduction of precipitation down wind. Instead, you're perhaps referring to orographic precipitation characteristics, such as rain shadow. I wouldn't introduce the foehn effect here, as it doesn't really apply.

*Thank you for raising this point out, we will modify the text to orographic precipitation.*

Section 3.4.1: Can you find a different name for the EPEs in this section? Up until now, you have defined EPEs as extreme precipitation events with a 95% or 98% threshold for the value of extreme. However, in this section, EPEs now mean 'percentage of cases where the other sites receive more precipitation than their site-specific EPE threshold'. It is then confusing to try and interpret the results – especially the relative wet/dry of the locations. However, this definition does answer a question I had earlier about whether thresholds were site-specific. With the current definition, I am left confused about whether IC is drier or wetter than other sites during EPEs.

*We understand that this is confusing, so we will replace the term "EPEs" in the table and text with "Above EPE threshold". We will also better explain what is done in this section so that the results are easier to interpret. We will rework the text to better highlight how we obtain the frequency distributions of precipitation anomalies and how the classification is made (i.e., that the "above EPE threshold" events correspond to EPEs too, meaning that EPEs are simultaneously occurring at the two sites).*

Whilst the differences in EPEs and negative anomalies per location from RACMO (table 4) seem significant, the differences between the sites in the downscaled data seems negligible or insignificant. Have you run any statistical tests on these results? The neg.anom for EPEs at IC and EPEs at TIR are very similar, and the two data sets do not agree with each other. The results here focus on the RACMO set, but you don't discuss the lack of consensus among the datasets. This could be because the downscaled data includes a longer time period, but as stated in your earlier results, there is no long-term trend in the data for precipitation, SMB or EPEs, so this perhaps doesn't answer it. I am really not convinced

with section 3.4.1 I understand your hypothesis and perhaps the method of trying to look at it, but the results are quite confusing.

*We understand this section is confusing, we will rewrite it to make it easier to understand. As stated previously, the downscaled data will be more discussed in the reworked version of the manuscript. We will include a discussion on the similarities and differences between the RACMO and downscaling outputs.*

Section 4: This first paragraph about ice core complexity should go in the introduction – throughout the results, I am concerned with how RACMO is representing observations, but this section gives me pause about the ice cores as observations. This level of complexity regarding SMB from ice cores should come earlier, especially for readers who are not experts in ice core interpretation.

*As said previously, we agree that this should come earlier in the manuscript. A brief overview of the SMB variability in ice core records is already given in the introduction (lines 70-74), but this will be further detailed in the revised version of the manuscript with the section dedicated to the comparison of the different SMB.*

If 2.2km RACMO was found to be more representative of the ice cores than 5.5km RACMO, why not use the higher resolution one? Or is the 2.2km RACMO the downscaled product you have used?

*Regarding the comparison with the ice core SMBs, the 2 km scale RACMO is closer than the one of the 5.5 km scale for IC and TIR, but not for FK. However, as the downscaling dataset is based on the 5.5 km scale, we made the choice to use this version throughout the paper, to analyse "comparable items". Moreover, the 5.5 km scale RACMO has been well evaluated in our region of interest (i.e., Lenaerts et al., 2014; Kausch et al., 2020; Ghilain et al., 2022) and we had easy access to the daily datasets because of previous projects we were involved in while the 2 km RACMO dataset has been only publicly archived at annual resolution. In addition, the 2 km RACMO results from the statistical downscaling of the 27 km scale RACMO with a high resolution surface topography at 2 km. RACMO 5.5 is thus expected to better represent the atmospheric and surface dynamics.The comparison we made between the 5.5 and 2 km scales aimed to recognize that the resolution used to analyze RACMO has a potential impact on the results. However, we realize that mentioning the 2 km RACMO dataset adds confusion in the manuscript. Instead of conducting the comparison in the discussion, we will propose the use of models with a finer resolution (both for RACMO and downscaling) as well as enhanced representation of processes, in particular processes related to the blowing snow, as a perspective to develop in the future.*

Minor:

Section 2.1: Can you provide the elevation of the ice rises – this becomes fairly important for your discussion on foehn winds and the loss of moisture across the trajectory.

*This is a good point, it will be added.*
*The IC12 core was drilled in December 2012 at the crest of DIR (−70.24218 °S, 26.34162 °E, ~429 m ASL) and is 120 m long. The FK17 core was drilled during the 2017/2018 austral summer at the crest of LIR (-70.53648° S, 24.07036° E, ~333 m ASL) and is 208 m long.*

*The TIR18 core was drilled during the 2018/2019 austral summer at the crest of HIR (-70.49960° S, 21.88017° E, ~348 m ASL) and is 262 m long.*

Line 165 and 168 say the same thing.

*The first sentence of line 165 will be deleted.*

Section 3.2.1 – is this really interannual variability section, or is it more spatial variability? Apart from the first line, the rest of this section is about the different locations.

*Thank you for pointing that out. We will merge the two sections and keep the section 3.2 title.*

Line 182: What do you mean by opposing signals in ice core records? Is this figure 2? Apart from TIR which has a decreasing trend, they don't seem to have opposing signals. This is hard to tell in figure 3 too.

*We will reference the text to Fig. 2. Opposite is indeed an inappropriate word. We will replace it by contrasting or different, as the three ice core records indicate different behaviors/trends.*

Table 3: caption says it is contribution to the total annual precipitation, which you also confirm in line 252, however in line 259 you say that EPE variance accounts for 2/3 of the SMB variance. So is the variance SMB and the average contribution annual precipitation? Different variables are used between RACMO (annual precipitation) and downscaled data (snowfall) – are they comparable?

*The caption of Table 3 mentions both contribution and variance, which are two distinct concepts. However, we will define variance, as a proper definition is lacking in the text. Regarding the different variables between RACMO and downscaled data, we addressed the question above.*

Line 327: I don't understand this sentence – where are the observed global atmospheric pathways observed?

*We meant that from Figure 6, it seems that the synoptic conditions are similar during EPEs at each of the three sites. But it also appears that some 95th EPEs at IC result in negative anomalies at the two other sites, thus we want to see if these correspond to particular synoptic patterns or not. We will change the text to make it clearer.*

Figure 8: ERA5 data?

*Indeed, these are ERA5 data, we will mention it.*

Line 356: change 'excludes' to 'rejects'.

*We agree.*

Line 438-439: Perhaps include that this is a conclusion from a model, not from observations.

*Thank you for pointing that out, it is important to remind readers that this comes from a model.*

---

## Author Response (AR2)

**Author response to the reviews - 2**

We would like to thank the two reviewers for their detailed and constructive comments, which greatly helped us to improve the manuscript.

According to the comment of Referee 1, we have modified sentence L264-265: "The 98th percentile corresponds to the largest 2% of daily precipitation events, of which approximately half are caused by the intense localized snowfall events produced by atmospheric rivers (Wille et al., 2021)."

As for comments from Referee 2, SMB is no longer mentioned in line 172.
Figures 5, 7 and S4 have been modified so that the circles indicating the position of the ice cores are now more visible (in dark red). Regarding the comment on lines 324-325, the reference to Carter et al. (2022) and the mention of remaining differences between RACMO forced by ERA-Interim and RACMO forced by ERA5 have been specified.
Finally, regarding the comments specifically related to RACMO: 1) The altitude given in RACMO is relatively close to the altitude measured in the field, keeping in mind that this is a mean altitude for the entire grid cell (about 25 km$^2$), as opposed to the altitude of the summits where the ice cores are drilled. 2) We analysed the time series of the eight adjacent grid cells for each of the three sites. The mean state differs between grid cells, but the variability is highly similar, with time-series that are still highly correlated (with average correlation coefficients of 0.66, 0.85, and 0.77, for IC, FK, and TIR, respectively). Furthermore, there is no temporal trend, and the average of the adjacent grid cells is relatively close to the grid cell corresponding to the ice core location: the differences are in the order of 10 to 50 mm, compared to differences between the model and the ice core records in the order of 200-265 mm (L154-156). This does not change the conclusion that RACMO indicates a lower SMB than the ice cores and does not explain the temporal trends observed in the ice cores.

---

## Author Response (AR3)

**Author response to the Editor**

*We would like to thank the editor for her valuable feedback. Our response is in italics.*

- line 28: "contributed about"

*This has been changed.*

- line 33-35: clarify this increase refers to the thermodynamic, and not dynamic, response

*According to the comment, we have now introduced the notion of thermodynamics in the sentence as follows: "For each degree of warming, a thermodynamically driven precipitation increase of 7 % should be observed".*

- line 43: I suggest rephrasing, thermodynamics is a field of study encompassing much more than temperature changes

*Indeed, we modified the sentence as: "Thermodynamics, including the effect of higher temperatures, is one of the three mechanisms controlling precipitation variability".*

- Section 2.2: I suggest swapping the paragraph order, first introducing RACMO and then describing how the model calculates SMB, to better fit the section title

*This has been modified accordingly. First, we introduce RACMO and its ability to simulate SMB and surface processes. Then, we define SMB and its components. Finally, we explain which SMB component of RACMO2.3 we will use.*

- for the hypothesis about orographic effects potentially underlying the spatial patterns, did the authors also look at the prevailing lower level wind direction?

*We looked at low-level winds, both in terms of strength and direction. We did not see any evidence of a foehn effect, probably because either the model is unable to represent the local effect of ice rises given its resolution, or either because the ice rises are a too small topographic feature to induce a significant effect on precipitation.*

- line 427: since which half of the 20th century?

*Since the second half of the $20^{th}$ century, this has been specified in the updated manuscript.*

- the manuscript essentially presents a null result, namely that spatial variability in precipitation as assessed by the RACMO and downscaling datasets does not explain the observed spatial variability in SMB. Therefore, the stated aim to "understand the variability observed in the SMB from three ice cores" is not achieved. I kindly ask the authors to adjust the text in certain places to further clarify this aspect of the paper (abstract, in the discussion, e.g. line 435).

*We have clarified the following passages in the text:*

*l.21-24: Shedding light on the intricate nature of SMB variability, our results also demonstrate that precipitation and EPEs alone cannot explain the spatial variability observed in the SMB records among the three ice core sites and suggest that other processes may be at play.*

*l.81-82: In this paper, we test the hypothesis that precipitation is the process driving the SMB spatiotemporal variability observed at the three ice rises mentioned above.*

*l.434-435: This study analyzes the spatial and temporal variability of precipitation and extreme precipitation events using the RACMO2.3 and downscaling datasets. This aims to test the hypothesis that the spatiotemporal variability observed in the SMB from three ice cores is driven by precipitation processes.*

*l.448-449: Overall, the spatial variability in modeled precipitation from the RACMO2.3 and downscaling datasets does not explain the observed variability in the three SMB ice-core records.*